# Cost Analysis Related to Diagnosis, Treatment and Management of Cervical Cancer in Antigua and Barbuda: A Prevalence-Based Cost-of-Illness Study

**DOI:** 10.3390/ijerph21121685

**Published:** 2024-12-18

**Authors:** Andre A. N. Bovell, Cebisile Ngcamphalala, Dane Abbott, Jabulani Ncayiyana, Themba G. Ginindza

**Affiliations:** 1Discipline of Public Health Medicine, School of Nursing and Public Health, University of KwaZulu-Natal, Durban 4000, South Africa; cn2323@cumc.columbia.edu (C.N.); ncayiyanaj@ukzn.ac.za (J.N.); ginindza@ukzn.ac.za (T.G.G.); 2Obstetrics and Gynaecology Department, Sir Lester Bird Medical Centre, St. John’s 35301, Antigua and Barbuda; dane.abbott@msjmc.org; 3Cancer & Infectious Diseases Epidemiology Research Unit (CIDERU), College of Health Sciences, University of KwaZulu-Natal, Durban 4000, South Africa

**Keywords:** Antigua and Barbuda, uterine cervical neoplasms, cervical cancer, cost of illness, economic burden, cost analysis

## Abstract

Cervical cancer remains a significant public health issue globally. In Antigua and Barbuda, cervical cancer is ranked among the top five most common cancers in terms of incidence and mortality among females. There is no evidence that the costs of diagnosing, treating, and managing this cancer have been studied before in Antigua and Barbuda. From the providers’ perspective, this study aimed to estimate costs associated with cervical cancer in Antigua and Barbuda. The prevalence-based cost-of-illness methodology was used to assess patient data abstracted from four study sites for the period 2017–2021, and to derive the annual prevalence. Top-down and bottom-up costing approaches were used to estimate direct medical costs. Costs were computed using the 2021 price level and converted to United States Dollars (USD). Total annual direct medical costs of cervical cancer were estimated at USD 0.24 million (ranging between USD 0.19 million and USD 0.30 million). Major cost drivers were treatment (USD 112,863.76), post-treatment side-effects care (USD 67,406.57), and the diagnostic process (USD 26,238.58). The overall direct medical unit costs for managing a case were estimated at USD 115,822.09. Our study reflects the current estimates for managing cervical cancer and provides evidence to complement cervical cancer prevention and cost containment measures in Antigua and Barbuda.

## 1. Introduction

The burden of cervical cancer continues to be a significant public health challenge worldwide [1]. Cervical cancer is one of the most common cancers in terms of incidence and mortality among women [1]. In 2020, cervical cancer accounted for about 604,000 new cases and 342,000 cancer deaths, with its burden disproportionately higher in low-income countries compared to high income countries [1].

In Antigua and Barbuda, cervical cancer is ranked among the most common cancers among women in terms of incidence and mortality [2,3]. Cervical cancer is known to have an infectious origin from the preventable human papilloma virus (HPV) [1]. Cervical cancer places a considerable burden on health systems given its direct and indirectly associated management costs (screening, diagnosis, management) [4]. Evidence shows that the costs associated with cervical cancer vary widely by country [5]. Ginindza et al., 2017, estimated the total annual direct medical costs associated with screening, managing, and treating cervical lesions, cervical cancer and genital warts in Eswatini, formerly Swaziland, at USD 16 million [6]. In an earlier study conducted in Malaysia, the cost of managing cervical cancer in the public setting was USD 75,888,329.45 [7], while in India, Singh et al., 2020, reported the cost associated with cervical cancer as ranging from INR 19,494 to 41,388 (USD 291–617) [8]. Further, in reporting on the economic burden of cervical cancer in Eswatini, Ngcamphalala and colleagues estimated direct costs to be USD 13.7 million [9], a value similar to the USD 12,589,360 estimated by Berraho et al., 2012, for Morocco [10]. Moreover, the findings of a real-world study showed that the aggregate direct medical costs of cervical cancer, which accounted for hospitalizations, appointments, and procedures carried out across three Latin American countries, namely, Brazil, Columbia and Mexico, varied considerably from USD 46,514,395.93 in Brazil to USD 52,862,760.88 and USD 30,174,473.64 in Columbia and Mexico, respectively [11], thus underscoring the vast disparity in the cost burden of cervical cancer across low to middle income countries.

Globally, the economic costs associated with cervical cancer are projected to rise [1]. Information on the economic burden of cervical cancer in Antigua and Barbuda is lacking. This study, therefore, aimed to estimate the economic burden of cervical cancer in Antigua and Barbuda from the healthcare provider’s perspective.

## 2. Materials and Methods

### 2.1. Study Area

Antigua and Barbuda is the largest of the English-speaking Leeward Islands with a population of 85,567 persons (53% females) and 80% of the population between the ages 15–74 [12]. It is a twin-island state, with Antigua, the larger one, located 650 kilometres southeast of Puerto Rico, and Barbuda, the smaller one, located 48 kilometres north of Antigua [10].

The country’s monetary policy is guided by the Eastern Caribbean Central Bank (ECCB), Basseterre, St. Kitts and Nevis, while fiscally, its main contributor to gross domestic product (GDP) is tourism [13].

Public healthcare is chiefly financed through contributions from the Antigua and Barbuda Medical Benefits Scheme, and Antigua and Barbuda Ministry of Finance allocations [14,15].

### 2.2. Study Population

The study population included histologically confirmed cases of women, aged 18 years and older, diagnosed with cancer of the cervix between 1 January 2017 and 31 December 2021. Cases were classified as per the International Classification of Diseases, 10th version (ICD-10) code (C53) [16].

### 2.3. Inclusion and Exclusion Criteria

There being no population-based or hospital-based cancer registry in Antigua and Barbuda, data on cervical cancer cases diagnosed between 1 January 2017 and 31 December 2021 were obtained retrospectively from medical records of the Sir Lester Bird Medical Centre (SLBMC), Antigua and Barbuda’s lone tertiary hospital, The Cancer Centre Eastern Caribbean (TCCEC), a public/private cancer facility (provided external beam radiation therapy plus other services), and the Medical Benefits Scheme (MBS), a statutory health organisation. Information on cervical cancer-related deaths was obtained from the Health Information Division of the Ministry of Health, Antigua and Barbuda.

Cases with precancerous lesions and recurrent disease were excluded from the study.

### 2.4. Diagnosis and Management of Cervical Cancer in Antigua and Barbuda (2017–2021)

In Antigua and Barbuda, most diagnosis and management of cancer is primarily conducted at the country’s lone tertiary care hospital, the Sir Lester Bird Medical Centre. Cancer management, including diagnosis and treatment of cancer of the cervix, is guided by treatment guidelines broadly adapted from World Health Organization’s (WHO) and the National Comprehensive Cancer Network Clinical Practice Guidelines (NCCN Guidelines) [17,18]. These guidelines provide guidance on optimising care and improving management outcomes with respect to the diagnostic process, and the staging and treatment of cancer patients [19]. Prior to September 2022, and during the period of 2017 to 2021, the Antigua and Barbuda public health system lacked an active and/or systematic screening programme for cervical cancer [20]. During the 2017 to 2021 period, the cervical cancer diagnostic process would have started with female patients either consulting a general practitioner or a gynaecologist directly based on presenting symptoms (such as spontaneous or contact bleeding, pelvic pain, abnormal vaginal discharge, dyspareunia) or opportunistically (for instance, if patient is sexually active or patient is ≥21years in age) [21]. At the gynaecological consultation, females would undergo a routine examination involving (i) physical assessment, (ii) clinical assessment, and (iii) a cytology smear (pap smear) which was the primary screening test used at the time. Detection is based on the initial findings of the physical examination in conjunction with the results of the cervical smear (Pap smear), all of which may or may not suggest the presence of abnormal cells in the cervix [21]. That is to say that the cells may be (i) negative for intraepithelial lesion (NLM), (ii) classified as atypical squamous cells of undetermined significance (ASCUS), a (iii) low-grade squamous intraepithelial lesion (LSIL), (iv) high-grade squamous intraepithelial lesion (HSIL), (v) abnormal glandular cell, or as (v) malignant. Should the initial findings indicate the presence of abnormal cells, then a cervical punch biopsy is taken at colposcopy for histopathologic assessment, and to determine whether cervical cells are precancerous (cervical intraepithelial neoplasia: CIN I-abnormal cells occupy one-third of the epithelium, CIN II-abnormal cells occupy two-thirds of the epithelium, CIN III-abnormal cells occupy the full length and thickness of epithelium) or cancerous lesions [22]. Depending on the severity of the lesion as seen on the Pap smear and based on the age of the patient, a histopathologic diagnosis is then sought through colposcopy and biopsy. If the biopsy is positive for CIN infection, then this is treated by a Gynaecologist based on the classification and in accordance with the American College of Obstetricians and Gynecologist’s (ACOG) guidelines [23]. Where the lesions are cancerous, then the biopsied specimen is further assessed to provide a histopathological stage for the disease. Pathological staging relies on the tumour–node–metastasis classification system (TNM) of the American Joint Committee on Cancer Staging System [24], and the International Federation of Gynaecology and Obstetrics (FIGO) classification system [21]. Under the AJCC system, the specimen is assessed to provide details on the size and spread of the tumour (T-primary tumour), how many lymph nodes are involved (if lymph nodes are in the sample) (N-regional lymph nodes), and whether there is pathological evidence of metastasis at a distant site (M-distant metastasis) [24]. With evidence of a positive biopsy for cancerous lesions (squamous cell carcinoma), the disease is then referred to a medical oncologist for clinical staging and management. Clinical staging entails the concomitant utilisation of the results of the physical examination with colposcopy and histopathology (cervical punch biopsy or conization), and is based on the International Federation of Gynaecology and Obstetrics (FIGO) classification system [21]. This involves the use of results from (i) a Complete Blood Workup; (ii) radiography (plain chest X-ray, abdominal/pelvic computed tomography scan with contrast, magnetic resonance imaging (MRI), electrocardiography); and (iii) endoscopy (cystoscopy, or sigmoidoscopy if bladder or colon invasion is suspected). Clinical staging occurs before any treatment and is used to determine whether the disease is localised (stage 1—disease is confined to the cervix uteri; stage 2—disease extends beyond the uterus but does not affect the lower third of the vagina or pelvic wall) or disseminated (stage 3—disease is in lower third of the vagina and may extend to the pelvic wall and beyond; stage 4—extends into the bladder or rectum or organs or bones) as determined by the tumour size, depth of cervical stromal invasion and disease extent (locoregional spread) [21]. Management is stage- and disease extent-dependent and may include surgical resection (hysterectomy usually for older females), concomitant radiation (external beam radiation therapy/brachytherapy—carried out overseas) and chemotherapy, or a combination of all [21] (Figure 1). The primary treatment for patients with localised disease (stages 1) may be surgery or radiation therapy, with or without systemic therapy, with a preferred cisplatin regimen or carboplatin where cisplatin tolerability is poor [18]. The primary treatment for patients with stage 2 disease is (i) surgery followed by external beam radiation therapy and systemic therapy (chemotherapy), in that order, with patients needing to travel overseas for brachytherapy or (ii) surgery followed by chemoradiation (chemotherapy and external beam radiation given locally, with brachytherapy being obtained overseas). Surgical oncological services are obtained through a visiting arrangement, with post-operative care being carried out by local gynaecologists and medical oncologists under guidance from the visiting oncology surgeon. In some cases, post-operative patients may directly travel overseas for chemoradiation (chemotherapy plus external beam radiation and brachytherapy). Treatment of advanced or disseminated cervical cancer (stages 2b, 3, 4a) involves the use of systemic (chemotherapy) and external beam radiation therapy and brachytherapy. Treatment of disseminated disease, stage 4b, is solely palliative, with patients being offered systemic therapy with or without external beam radiation [18]. Further, given peculiarities that may exist for presenting patients, if a person presents with signs and symptoms of higher stage disease, then screening would not be carried out. A tissue diagnosis is usually obtained, clinical diagnosis is made, and then the patient commences treatment.

Fertility sparing surgery, brachytherapy and positron emission tomography (PET scan) are currently not available in Antigua and Barbuda [25]. Cervical cancer patients requiring these services are referred to other countries for treatment. Except for fertility-preserving surgery, the Medical Benefits Scheme covers all cervical cancer-related costs for treatment abroad.

In this study, cost estimates for all care components involved in the diagnosis and management of cervical cancer were based on 2021 market prices as per sources expressed in terms of private clinics and laboratories, and tertiary hospital unsubsidized prices (Table 1).

### 2.5. Cost-of-Illness

This was a prevalence-based cost-of-illness (COI) study conducted from the healthcare provider’s perspective [26]. Use of the cost-of-illness method was premised on its advantage of being a simple and easy tool for identifying the drivers of diagnosis and treatment costs associated with several diseases [26]. Cost-of-illness studies may be described as being prevalence-based or incidence-based [26], with prevalence-based studies used to estimate the economic burden or total costs of a condition over a specified period (usually a year), and incidence-based studies used in estimating the lifetime costs of an illness from diagnosis until its disappearance or another endpoint [26].

For conditions that are profoundly burdensome on small countries, such as those in the English-speaking Caribbean, with their budgetary restrictions and relatively disadvantaged health system, cost studies using the cost-of-illness methodology have the potential to empower public health policymakers and healthcare providers with information on specific diseases or conditions [27]. Usage of the COI methodology, therefore, is an enabler for the identification, quantification, and valuation of resources related to any named condition from several perspectives, inclusive of those of international, governmental, societal, and healthcare providers [9,27]. This study therefore made ample use of the prevalence based cost-of-illness method in estimating economic burden of cervical cancer, a chronic disease that weighs heavily on health expenditure in Antigua and Barbuda [27,28].

Healthcare interventions required in the cervical cancer management continuum (diagnosis and management) were identified (micro-costing approach), quantified and valued per case and extrapolated using prevalence data to estimate national costs. Data were collected retrospectively using patient charts and excel spreadsheets.
Direct Medical Costs of Disease (mc) = ∑(Mi × Pi)(1)
Mi is number of cases requiring healthcare;Pi is the required healthcare resources unit costs per case;mc is the total costs.

We investigated only direct medical costs, given their usefulness in evaluating disease burden in a healthcare setting [28,29] and as determined using a top-down and bottom-up approach [26]. Direct medical costs considered recurrent costs [6]. Recurrent costs included personnel, travel, consumables supplies (medical and non-medical), treatment, administration, and overheads [6].

For our cost analysis, we estimated the number of cervical cancer cases requiring healthcare in a single year in Antigua and Barbuda, computed by subtracting the number of deaths from the total number of cases, with the answer being divided by 5 [6]. For comparative purposes, and to assess, directly, the impact of unrecorded cases at our study sites, the average number of cancer cases in a single year was increased by 50% and the total annual costs were assessed and reported [6,28]. The direct medical unit costs of care components were also increased by 50%, and the total annual costs were assessed and reported [6]. Additionally, treatment cost in the original model was reduced by 50%, while the costs of other parameters remained unchanged, and the impact of this manipulation on the total annual costs was assessed and reported.

### 2.6. Cost Data

Direct medical costs related to diagnosis, treatment, and follow-up care for cervical cancer were collected using an electronic datafile (excel spreadsheet) designed based on available on-site datafiles and routine records available at our study sites. This data collection tool was based on the cervical cancer diagnostic process and general treatment pathway employed in Antigua and Barbuda (Figure 1).

Direct medical costs were obtained from reimbursement receipts submitted to the Medical Benefits Scheme as part of the financial claims applications for laboratory and pathology testing, imaging, surgery, ultrasound, hospitalisation, medication, and other similar services provided to beneficiaries by private health facilities on the island [15,30]. Additional cost information was obtained from private health facilities including clinics and laboratories.

Direct medical costs were computed by multiplying the number of cases requiring healthcare by the corresponding costs of each resource per case. Total costs represented an aggregate of the costs of all care parameters.
Direct Medical Costs of Disease (mc) = ∑(Mi x Pi)(2)

Costs were reported in 2021 USD. This was attained following adjustment of the country’s consumer price index (CPI) of 2021 and the USD 2021 exchange rate (USD 1 = XCD 2.7169) as below:*Value in* 2021 USD = *base year price* × CPI in 2021/CPI *in based year*(3)

CPI in 2021 = 95.27; CPI in based year = 95.27 [31].

Further, given that brachytherapy and PET scans, including their associated transportation and accommodation costs, constituted significant healthcare imports financed by the Medical Benefits Scheme for cervical cancer patients, total costs were presented with and without the inclusion these care components [32].

### 2.7. Sensitivity Analysis

Using a method that was applied and mentioned in other studies, we performed sensitivity analysis using the lower and upper bound of ±25% so as to account for the impact of uncertainty in the cost estimation and any unrecorded cases by the study sites used in our cost-of-illness study [6,9].

### 2.8. Ethical Considerations

Approval for this study was granted by the Antigua and Barbuda Institutional Review Board, Ministry of Health (AL-04/052022-ANUIRB), the Institutional Review Board of the Sir Lester Bird Medical Centre, and the University of KwaZulu-Natal Biomedical Research Ethics Committee (BREC/00004531/2022). Data on our population of cervical cancer cases for the study period 2017–2021 was accessed only by the principal investigator, who is also the corresponding author, from 16 September 2022 to 16 January 2023, at the study sites. Additionally, all cost data based on 2021 prices were collected by the same author during the period of 22 November 2022 to 25 January 2024. This study did not involve direct contact with cases, and there was no direct risk to persons [33]. De-identification and anonymization were ensured by not recording the names of any of the patients at, during, or after data collection [33].

## 3. Results

### 3.1. Background Demographic Information

Between 2017 and 2021, there were 40 women diagnosed with cervical cancer in Antigua and Barbuda and the average age was 52 years (Table 2). About 13% of the women diagnosed were between ages of 20–34 years, whilst approximately 50% and 30% were between the ages of 35–54 and 55–74 years old, respectively. Only 45% of the cases were diagnosed with early stages of the disease (clinical stages I–II) (Table 2).

### 3.2. Estimate of Cervical Cancer Cases in a Single Year

Information obtained from our study sites indicated that 14 of the 40 diagnosed individuals had died during the study period. We therefore estimated that there were five cases of cancer on average in a single year in Antigua and Barbuda. This was calculated by subtracting the 14 deaths from the 40 cases diagnosed in the period and dividing the result (26) by 5.

When the average number of cases in a single year (five) increased by 50%, the estimated number of cases per year changed to eight.

### 3.3. Total Unit Costs

The overall costs for diagnosing, treating, and managing a cervical cancer patient was approximately USD 115,822.09 (USD 107,019.68) when costs for the healthcare imports were excluded) (Table 3). The main cost drivers included post-treatment side-effects relieving care accounting for 40%, (USD 46,669.65), treatment and follow-up care accounting for 35% (USD 40,418.19) and 17% (USD 19,581.14), respectively, (Figure 2A). Among treatment components of the overall costs, radiotherapy accounted for 46% (USD 18,838.11) of cost (Figure 2B). The leading cost drivers for post-treatment side-effects relieving procedures were renal complication/renal failure which accounted for 44% (USD 20,674.86) of costs, followed by other post-treatment complications which accounted for 30% (USD 13,942.77) of the total costs (Figure 2C).

The estimated direct medical unit cost of cervical cancer by FIGO stage ranged from USD 60,434.07 (adjusted—where the costs of healthcare imports were included in the total) or USD 57,495.42 (unadjusted—where the costs of healthcare imports were excluded from the total) when surgery was considered as the preferred treatment option in managing stage I-Ib/IIa disease. When the primary treatment was radiation therapy or radiation and systemic therapy, the adjusted cost was USD 73,567.15 and USD 76,586.41, respectively.

For stage IIB-II, costs were estimated to be USD 97,261.27 (adjusted) and USD 88,458.85 (unadjusted); for FIGO stages III and IV, costs ranged from USD 108,588.11 (adjusted) or USD 99,785.69 (unadjusted) for stage III, to USD 89,750.00 (adjusted) or USD 86,811.35 (unadjusted) for stage 4 disease. (Table 3). Across all cost parameters per case, diagnostic process and other direct costs were the cheapest, accounting for 5% (USD 5214.59) and 3% (USD 3938.52) of overall direct medical unit costs, respectively. Interventions that were major cost drivers included radiotherapy, specifically External Beam Radiotherapy (USD 12,974.35), and targeted therapy or special chemotherapy drugs for disseminated disease (USD 12,855.79). Other significant drivers were side-effect-relieving procedures, specifically regarding renal complications/renal failure, and other complications, recorded at USD 20,674.86 and USD 13,942.77, respectively (Table 3).

### 3.4. Total Annual Direct Medical Costs

The total annual direct medical costs for cervical cancer were estimated (sensitivity analysis (±25)) to be USD 238,439.76 (ranging between USD 178,829.82 and USD 298,049.70), Table 4. When healthcare imports were excluded from the analysis, this value was estimated at USD 206,155.23 (range of USD 154,616.42 to 257,694.04) (Table 4). The top three contributors to total annual direct medical costs included treatment, at USD 112,863.76 (ranging between USD 84,647.82 and USD 141,079.70), followed by post-treatment side-effects care, at USD 67,406.57 (ranging between USD 50,554.93 and USD 84,258.21), and the diagnostic process at USD 26,238.58 (ranging between USD 19,678.94 and USD 32,798.23) (Table 4) (Figure 3). Prior to including the costs of healthcare imports, other direct costs were estimated at USD 4999.30 (Table 4).

When the total annual direct medical costs were adjusted for the use of radiotherapy and chemotherapy in the treatment of early-stage disease, the annual direct medical costs increased slightly to approximately USD 254,592.10 (ranging between USD 190,944.08 and USD 318,240.13) (Table 4).

Additionally, to investigate the impact of how treatment costs affected the results, we decreased the direct medical unit costs for each cancer stage by 50%, and the total annual direct medical costs were estimated at USD182,007.88 (ranging between USD 136,505.91 and USD 227,509.85).

After increasing the direct medical unit costs of our care components by 50%, the total annual cost estimate then changed to USD 357,659.64 (ranging between USD 268,244.73 and USD 447,074.55), representing a corresponding 50% increase in overall total annual direct medical costs.

Following the increase in the average number of prevalent cases by 50% (from five to eight for a single year), the estimated total annual direct medical cost changed to USD 366,251.56 (ranging between USD 274,688.67 and USD 457,814.45), which represented a 54% increase in total annual direct medical costs (Table 5). This suggested that each additional prevalent case over and above five cases would be responsible for USD 42,603.93 being added to total annual cost. With the exclusion of healthcare imports, our estimated value was reduced to USD 313,423.56 (ranging between USD 235,067.67 and USD 391,779.45), with the cost drivers of treatment and other direct costs also showing a corresponding decline in their estimated values as well (Figure 3).

## 4. Discussion

This study reported the economic burden of cervical cancer in Antigua and Barbuda by estimating the direct medical costs from the healthcare provider’s perspective. Our study showed that the estimated total annual direct medical cost of cervical cancer in 2021 was USD 238,439.76 (ranging between USD 178,829.82 and USD 298,049.70), displaying a 14% reduction when healthcare imports of brachytherapy and PET scans and their associated transportation and accommodation costs were removed from the analysis (USD 206,155.23: range of USD 154,616.42 to USD 257,694.04). The major cost drivers of the total annual direct medical cost were treatment, post-treatment side-effects care, and the diagnostic process, which accounted for 47% (USD112,863.76), 28% (USD67,406.57), and 11% (USD 26,238.58) of the total annual costs, respectively. The study also demonstrated that, ceteris paribus*,* an increase in average prevalent cases from five to eight would result in the total annual direct medical costs increasing to USD 366,251.56 (range of USD 274,688.67 to USD 457,814.45), with suggestions of a USD 42,603.93 increase in this estimate for every additional prevalent case included in our model. Similarly to observations made in our model with five prevalent cases, treatment (51% or USD 185,309.33), post-treatment side-effects (24% or USD87,970.52) and the diagnostic process (11% or USD 41,882.35) retained their positions as the major cost drivers. Observably, this affirmed treatment and post-treatment side-effects care as serious cost drivers of cervical cancer care in Antigua and Barbuda.

Most studies have shown that the economic burdens of cervical cancer vary across countries and, in part, this could be attributed to differences in the disease burden [6,34,35]. Although our estimates of annual direct medical cost appear to be lower than those of high income countries, they may yet represent a significant economic burden when considered in the context of Antigua and Barbuda’s small population size, existing health system, the budgeted national health allocations for 2021, and current health expenditure per capita, which was USD 923.41 in 2021 [36,37].

Many studies on the economic burden of cervical cancer conducted in other countries have often used different methodological strategies to estimate and present costs [28,38]. Fully understanding our main cost drivers in this context could provide information on cost containment discussions either at a health facility or policy level in Antigua and Barbuda. In this regard, and notwithstanding our choice of cost-of-illness method, the findings regarding the treatment parameter being the largest cost driver seems consistent with that of other studies which have shown that the costs of cervical cancer vary greatly by country. For example, in Eswatini, a low- to middle-income country, the annual costs of cervical cancer were estimated at USD 16 million, compared to about USD 7.5 million (EUR 6.7 million) in Bulgaria, an upper middle income country, and USD 94 million as reported by Sweden, a high-income country [6,34,35].

The overall unit cost (inclusive of the diagnostic process and treatment per patient) was estimated at USD 115,822.09. Within the treatment category, radiation therapy (external beam radiation, USD 12,974.35, plus brachytherapy, USD 5863.76)) was the major cost driver, followed by chemotherapy (targeted therapy, USD 12,855.79, plus cisplatin-based systemic therapy, USD 3019.26) and surgery (USD 5705.03). The high costs related to chemotherapy and radiation therapy agreed with the findings of other studies which show that chemotherapy and radiotherapy tend to be the main cost drivers of treatment [34,39]. The low cost for surgery could be attributed to the exclusion of trachelectomy as a fertility-preserving option for women of childbearing age, and the fact that the median age at presentation for most women was 52 years, with suggestions of patients for surgery being excellent candidates for the low-cost option of hysterectomy because of their reproductive status. Given its absence in the local treatment pathway, research that considers the impact of this surgery type on direct medical costs could be considered in the future.

Similarly, finding renal complications/renal failure to be the major cost driver for post-treatment side-effect-relieving procedures, accounting for about 44% (USD 20,674.86) was uniquely different from evidence shown in previous studies. This observation could in part be associated with the effect of cisplatin-based concurrent chemoradiation more likely being linked with sub-optimal pre-hydration, especially for already at-risk patients [40]. Rose et al., 1999, argued that concurrent use of chemotherapy and radiation is the most common intervention for patients undergoing cervical cancer treatment, given its advantage of increasing the sensitivity of cancer cells to radiation [41].

The diagnostic process, which consisted of several relatively low cost care components that are necessary in cervical cancer screening practises, especially in low-resource settings, underscores the need for and feasibility of Antigua and Barbuda investing more in screenings, which could lead to higher cervical cancer detection and curation efforts [42], especially given the country’s recent effort to reduce the incidence of cervical cancer in the foreseeable future [43].

Compared to the disease costs by cancer stage, the cost of stage I-IB/IIA was found to be lower than those of the other clinical stages of the disease. There was a noted progression of costs, with the highest being that of the stage III disease, followed by a lowering of the cost of the stage IV disease. While costs by clinical stages have been found to vary in previous studies, our findings are consistent with the results of studies indicating that early-stage treatment usually presents with lower costs compared to late-stage treatment, thus affirming that cost is generally based on the severity of disease [10,44,45]. In the Antigua and Barbuda situation, progression in costs is likely to be attributed to the severity of the disease process and its corresponding diagnosis, treatment, and side-effect management costs. Similar observations were noted by Berraho et al. when estimating the direct cost of cervical cancer management in Morocco [10].

Our findings suggest that the economic burden of cervical cancer from the healthcare provider perspective, though reasonable, and based on an algorithm that easily presents the preferred local approach to managing this disease, did not address gains from interventions, as would be the case if we had conducted a cost-effectiveness or cost–utility study. In the absence of local evidence to suggest that our approach to evaluating the economic burden of cervical cancer had been tried previously, our study results could be a useful benchmark for establishing the costs for care components important to the diagnostic process, treatment, and management of cervical cancer. Further, given that HPV infection is implicated in almost 100% of cervical cancer cases globally, and since treatment of cervical cancer is FIGO stage-dependent [46], with higher stage diagnoses suggestive of more severe disease [46], our study results are useful for evaluating the effectiveness of health policies aimed at harmonising the primary prevention tools of HPV vaccination and testing, along with secondary prevention methods of screening, detecting, and treating preclinical lesions in the fight against cervical cancer in Antigua and Barbuda [46]. Future studies could look at this by addressing the costs of cervical cancer from other perspectives, while providing advocacy for the implementation of cost containment strategies and resource allocation guidelines that could improve cancer prevention, detection, and survivability on the island.

Our study in presenting unique evidence on the direct medical costs associated with cervical cancer in Antigua and Barbuda also highlights the absence of a national cancer registry, a national register of costs, and legislation mandating the reporting of cancer cases to a national body in Antigua and Barbuda [25]. Having access to such services could help in improving the ascertainment and understanding of the distribution of prevalent cases of cervical cancer by accounting for all cases diagnosed and treated locally. Another strength is that the study was restricted to the three key study sites (sites providing the largest catchment of documented evidence on cancers in the country), mentioned in the Methods section, and depended on data obtained through a robust data collation process, involving discussions had with experts, including a pathologist, a medical oncologist, gynaecologists, a pharmacist, and oncology nurses at Sir Lester Bird Medical Centre, Antigua and Barbuda, as well as consultation of the literature to gain an understanding of the data in relation to the components of cervical cancer care. This approach ensured that, in addition to using a health economics method of costing, our receipt of expert guidance, along with our literature consultation, would provide the important and contextual support upon which our cost model is based. Thus, we are confident that our costs are sufficiently detailed and reflect the current economic burden of cervical cancer in Antigua and Barbuda.

Our study is not without limitations. Our study utilised a prevalence-based cost-of-illness approach from the provider’s perspective. This approach is not dynamic, and as such did not consider certain elements, such as the patient’s quality of medical care, quality of life, loss of productivity, and end-of-life care, and thus, intrinsically, it provided a constrained view of the economic burden of cervical cancer in Antigua and Barbuda [26]. Future cost models that incorporate these elements and consider cost-of-illness studies from other perspectives would be useful [26]. Another limitation is that our study used retrospectively collected data, which could have been affected by a priori recording or reporting biases, especially if there were inaccuracies in the recording or reporting of certain patient information. Further, our study’s scope did not allow for consideration of the costs of screening, treating, or managing precancerous lesions or precursor lesions of cervical cancer, nor did our estimates report on them. Compared to other studies that included costs for other HPV-related diseases such as CIN, squamous intraepithelial lesions (LSIL and HSIL), and adenocarcinoma in situ (AIS) [6], our presented results could be considered a conservative estimate of the economic burden of cervical cancer. Further studies that may cater to incorporating screening and treatment costs for precursor lesions, comparing screening costs for these lesions versus their treatment costs, while also examining positivity rate on cancer outcomes, are needed to determine the burden of other HPV-related cervical conditions as well as provide a more comprehensive estimate of the economic burden of cervical cancer in Antigua and Barbuda [43,46]. The absence of index costs or a national register of costs in Antigua and Barbuda was another of the study’s limitations. We considered private and market prices for the best possible price estimates which may vary from time to time. Additionally, information from the literature and interviews with experts to inform some of the treatment variables were considered in determining local diagnostic, treatment, and management pathways. Valuable as this was to our study, our reliance on experts, including their estimations, could easily have introduced biases toward presenting context and preferred management methodologies, especially if the experts consulted were prejudiced to one procedure over another, and/or utilised approaches to care that corresponded closely to those found in the countries in which they were trained. The effect of bias was mitigated by having our collated data and cost estimates validated and by accounting for model uncertainty with the inclusion of sensitivity analysis. Future studies that consider a pragmatic framework for assessing the economic burden of cervical cancer in Antigua and Barbuda could be performed [28].

Considering the above limitations, including our small number of prevalent cases and their effect on the true costs related to cervical cancer in Antigua and Barbuda, future analysis of costs, as could be obtained in regard to cost-of-illness from the societal perspective, could provide a broader view of the economic burden of this condition in the country.

## 5. Conclusions

Cervical cancer is one of the most serious gynaecological conditions affecting women in Antigua and Barbuda. The results of this study indicated that the cost associated with cervical cancer management in Antigua and Barbuda appear reasonable. The major cost drivers are treatment, post-treatment side-effects care, and the diagnostic process, with a gradation of costs by stage except in the case of stage IV disease, which showed a nominal decline in costs. Although we are confident that our cost estimates are sufficiently detailed and present a fair representation of the direct medical costs of cervical cancer in Antigua and Barbuda, we advocate that having an established national cancer registry and national register of costs to support future research which evaluates the economic burden of cervical cancer using the cost-of-illness method from the societal perspective, or through use of cost-effectiveness and/or cost–utility studies would be useful. Further, and aside from aiding health resources allocation guidelines and budget planning, our findings provide a useful benchmark for health policy makers to assess the effectiveness of strategies such as HPV vaccination, HPV testing, screenings, drug formulation management and medical supply procurement practises on the costs of cervical cancer in the foreseeable future.

## Figures and Tables

**Figure 1 ijerph-21-01685-f001:**
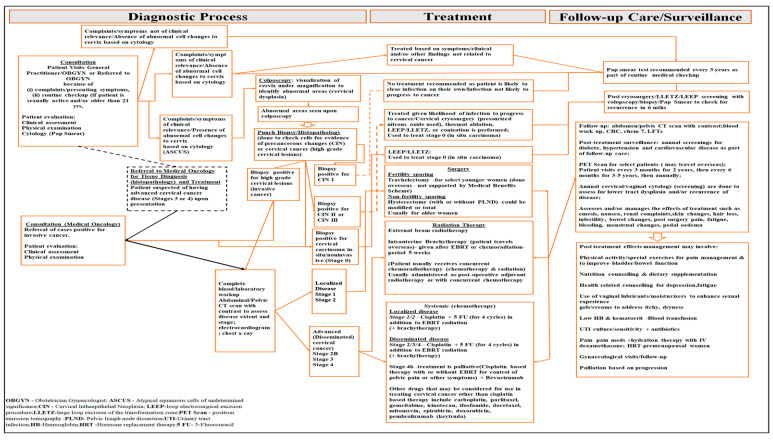
Schematic diagram showing the main components of care related to the management of cervical cancer in Antigua and Barbuda (2017–2021).

**Figure 2 ijerph-21-01685-f002:**
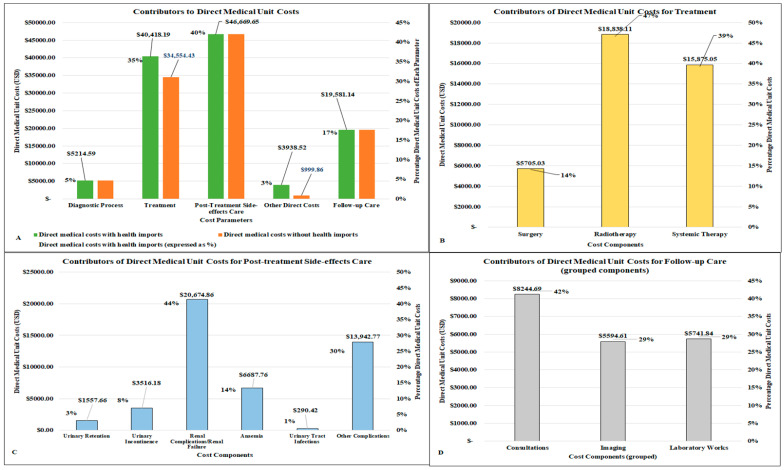
(**A**–**D**): Key contributors to direct medical unit costs and disaggregated costs of its main cost drivers.

**Figure 3 ijerph-21-01685-f003:**
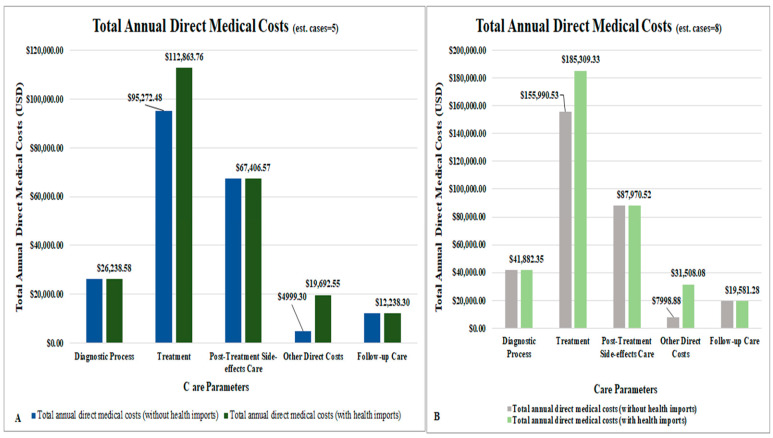
(**A**,**B**): Total annual direct medical costs broken down by care parameters and estimated number of cases in a single year.

**Table 1 ijerph-21-01685-t001:** Data variables and sources of costs regarding screening and the treatment and management of cervical cancer.

Data/Parameter	Data Explained	Data Source	Price Source
**Estimated Number of Cervical Cancer Cases**		**Combined Number of Cases Abstracted from Patient Files of the Eastern Caribbean Cancer Centre, the Sir Lester Bird Medical Centre, and the Medical Benefits Scheme for the Period 2017–2021**	**N/A**
**Diagnostic Process**			
Consultation	Patient is exposed to full clinical assessment and cytology smear	Interview with expert, Medical Benefits Scheme (review of reimbursement receipts presented for refunds and board-approved financing of care), Sir Lester Bird Medical Centre (billing and finance departments)	Market price
Clinical Assessment			Market price
Physical Examination			Market price
Cytology (Pap) Smear		Medical Benefits Scheme (review of reimbursement receipts presented for refunds and board-approved financing of care), Antigua and Barbuda private laboratories charges	Market price
Colposcopy	Colposcopy Cervical punch biopsy Computed tomography (CT scan) of the abdominal/pelvis; chest X-ray; Electrocardiogram) Complete Blood Workup following biopsy. Cytohistology (includes transportation and reporting on specimen)	Interview with experts, Medical Benefits Scheme (review of reimbursement receipts presented for refunds and board-approved financing of care), Sir Lester Bird Medical Centre (billing and finance departments), Sir Lester Bird Medical Centre Laboratory, Antigua and Barbuda private laboratories/clinic charges	Market price
Cervical Punch Biopsy			Market price
Laboratory (Complete Blood Workup)			Market price
Cytopathology			Market price
Imaging Studies (Computed Tomography (CT scan); X-ray; Electrocardiogram (ECG/EKG)			Market price
**Treatment**			
Surgery	Hysterectomy (offered locally); External Beam Radiotherapy, offered through local cancer centre; complementary to surgery, brachytherapy or chemotherapy. Brachytherapy offered overseas; complementary to surgery, brachytherapy or chemotherapy. Mainly Cisplatin + 5 Fluorouracil for 4 cycles (complementary to EBRT and brachytherapy), may include optional therapies for disseminated disease cases, such as Carboplatin + Paclitaxel and/or Bevacizumab and Pembrolizumab (palliation).	Interviews with consultant gynaecologist, medical oncologist, senior oncology nurses at Eastern Caribbean Cancer Centre and the Sir Lester Bird Medical Centre, interviews with pharmacists, Medical Benefits Scheme (review of reimbursement receipts presented for refunds and board-approved financing of care), Sir Lester Bird Medical Centre (billing and finance departments), general discussion with surgeon in private surgery, interview with manager of operating room at Sir Lester Bird Medical Centre.	Market price
Radiotherapy 1 (External Beam Radiotherapy)			Market price
Radiotherapy 2 (Brachytherapy)			MBS Approved Financing/Market price
Systemic therapy (Chemotherapy)			Market price/ Private Supplier
**Post-Treatment Side-Effect-Relieving Procedures**			
Urinary Retention	Any procedure or approach required by a patient that received cervical cancer related treatment; may involve responses to urinary retention or incontinence, renal complications/renal failure, or other complications such as depression, fatigue, etc.	Interviews with consultant gynaecologist, senior oncology nurses at Eastern Caribbean Cancer Centre and the Sir Lester Bird Medical Centre, interviews with pharmacists, Medical Benefits Scheme (review of reimbursement receipts presented for refunds and board-approved financing of care), Sir Lester Bird Medical Centre (billing and finance departments), general discussion with surgeon in private surgery	Market price
Urinary Incontinence			Market price
Renal Complications/Renal Failure			Market price
Anaemia (Low Haemoglobin/Hematocrit)			Market price
Urinary Tract Infections			Market price
Other Complications of Treatment (Post-Operative Pain, Bowel Issues, etc.)			Market price
**Other direct costs**			
Nutrition Counselling	Most services are available to all patients locally; patients do have access PET scan services overseas whenever it is required.	Interviews with consultant gynaecologist, senior oncology nurses at Eastern Caribbean Cancer Centre and the Sir Lester Bird Medical Centre, interviews with pharmacists, interview with nutritionist, Medical Benefits Scheme (review of reimbursement receipts presented for refunds and board-approved financing of care), Sir Lester Bird Medical Centre (billing and finance departments).	Private: Market price
Psychiatric/Psychological Counselling			Market price
Pharmacy Services			Private Pharmacy/market price
Imaging (PET Scan—Overseas)			MBS-approved financing/market price
Treatment (Brachytherapy—Overseas)			MBS-approved financing/market price
Emergency Kit			
Transportation/Accommodation (Imaging/Treatment—Overseas)			MBS-approved financing/market price
Transportation (Local Services Related)			Market price
Overheads			Market price
**Follow-up Care**			
Post-Treatment Consultations	Post-treatment Consultations; cytology smear, CBC w/ diff, renal panel tests, liver function tests, imaging studies. Follow-up care is performed to assess for both the effects of treatment and to monitor for disease recurrence; usually every 3 months for 2 years, then every 6 months for 3–5 years, then annually thereafter; requires regular cytology smears, along with a battery of laboratory tests and/or imaging studies	Interviews with consultant gynaecologist, medical oncologist, senior oncology nurses at eastern Caribbean Cancer Centre and the Sir Lester Bird Medical Centre; Medical Benefits Scheme (review of reimbursement receipts presented for refunds and board approved financing of care), Sir Lester Bird Medical Centre (billing and finance departments).	Market price
Computed Tomography (CT scan with contrast)			Market price
Complete Blood Count with Differential			Market price
Chemistry Panel 7			Market price
Renal Panel 1			Market price
Liver Function Tests			Market price
Cytology Smear			Market price
Glycated Haemoglobin (HbA1c)			Market price
Total Cholesterol			Market price

**Table 2 ijerph-21-01685-t002:** Background characteristics of the population of cases of cervical cancer (2017–2021).

Variable Class	Characteristics	Cervical Cancer N = 40, N (%)
**Demographic**	**Age at Presentation**	
	Mean age (SD)	51.8 (15.0)
	Mean age 95% CI	47.0–56.6
	Median age (IQR)	51.5 (16.0)
	Age Range	30.0–86.0
	**Age Distribution**	
	20–34	5 (12.5)
	35–54	20 (50.0)
	55–74	12 (30.0)
	≥75	3 (7.5)
**Clinical**	**Clinical Stage**	
	I	8 (20.0)
	II	10 (25.0)
	III	15 (37.5)
	IV	7 (17.5)

**Table 3 ijerph-21-01685-t003:** Direct medical unit costs for diagnostic processing, stage, treatment, and management of cervical cancer—stages I–IV.

Screening, Diagnosis, Treatment, and Management Variables	Overall Unit Costs (USD)			Units Costs by FIGO Stage (USD)			
**Estimated Number of Cases in a Single year (N = 5)**		**I-IB/IIA (n = 1) ^a^**	**I-IB/IIA (n** = **1) ^b^**	**I-IB/IIA** **(n = 1) ^c^**	**IIB-II (n = 1)**	**III (n = 2)**	**IV (n = 1)**
**Diagnostic process**							
**Consultation (Gynaecology) (Clinical Assessment, Physical Examination)**	USD 110.42	USD 110.42	USD 110.42	USD 110.42	USD 110.42	USD 110.42	USD 110.42
**Consultation (Medical Oncology)**	92.02	92.02	92.02	92.02	92.02	92.02	92.02
**Cytology (Pap) Smear**	USD 55.21	USD 55.21	USD 55.21	USD 55.21	USD 55.21		
**Colposcopy**	USD 1473.74	USD 1473.74	USD 1473.74	USD 1473.74	USD 1473.74	USD -	USD -
**Biopsy**	USD 1583.98	USD 1583.98	USD 1583.98	USD 1583.98	USD 1583.98	USD 1583.98	USD 1583.98
**Imaging (Radiology)**	USD 1012.18	USD 1012.18	USD 1012.18	USD 1012.18	USD 1012.18	USD 1012.18	USD 1012.18
**Laboratory**	USD 408.55	USD 408.55	USD 408.55	USD 408.55	USD 408.55	USD 408.55	USD 408.55
**Cytopathology**	USD 478.49	USD 478.49	USD 478.49	USD 478.49	USD 478.49	USD 478.49	USD 478.49
**Treatment**		a	b	c			
**Surgery**	USD 5705.03	USD 5705.03	USD -	USD -	USD -	USD -	USD -
**Radiotherapy 1 (External Beam Radiotherapy-EBRT)**	USD 12,974.35	USD -	USD 12,974.35	USD 12,974.35	USD 12,974.35	USD 12,974.35	USD -
**Radiotherapy 2 (Brachytherapy) ^m^**	USD 5863.76	USD -	USD 5863.76	USD 5863.76	USD 5863.76	USD 5863.76	USD -
**Systemic therapy (Chemotherapy) ****	USD 3019.26	USD -	USD -	USD 3019.26	USD 3019.26	USD 3019.26	USD 3019.26
**Targeted therapy *****	USD 12,855.79	USD -	USD -	USD -	USD -	USD 12,855.79	USD 12,855.79
**Post-Treatment Side-Effect-Relieving procedures**							
**Urinary Retention**	USD 1557.66	USD 1557.66	USD 1557.66	USD 1557.66	USD 1557.66	USD 1557.66	USD 1557.66
**Urinary Incontinence**	USD 3516.18	USD 3516.18	USD 3516.18	USD 3516.18	USD 3516.18	USD 3516.18	USD 3516.18
**Renal Complications/Renal Failure**	USD 20,674.86	USD -	USD -	USD -	USD 20,674.86	USD 20,674.86	USD 20,674.86
**Anaemia (Low Haemoglobin/Haematocrit)**	USD 6687.76	USD 6687.76	USD 6687.76	USD 6687.76	USD 6687.76	USD 6687.76	USD 6687.76
**Urinary Tract Infections**	USD 290.42	USD 290.42	USD 290.42	USD 290.42	USD 290.42	USD 290.42	USD 290.42
**Other Complications of Treatment (Post-Operative Pain, Bowel Issues, etc.)**	USD 13,942.77	USD 13,942.77	USD 13,942.77	USD 13,942.77	USD 13,942.77	USD 13,942.77	USD 13,942.77
**Other Direct Costs**							
**Nutrition Counselling**	USD 100.00	USD 100.00	USD 100.00	USD 100.00	USD 100.00	USD 100.00	USD 100.00
**Psychiatric/Psychological Counselling**	USD 128.82	USD 128.82	USD 128.82	USD 128.82	USD 128.82	USD 128.82	USD 128.82
**Pharmacy Services**	USD 59.99	USD 59.99	USD 59.99	USD 59.99	USD 59.99	USD 59.99	USD 59.99
**Positron Emission Tomography (PET) Scan (Overseas) ^m^**	USD 1540.00	USD 1540.00	USD 1540.00	USD 1540.00	USD 1540.00	USD 1540.00	USD 1540.00
**Emergency Kit (Chemo)**	USD 112.94	USD 112.94	USD 112.94	USD 112.94	USD 112.94	USD 112.94	USD 112.94
**Patient Transportation/Accommodation (Overseas Imaging) ^m^**	USD 1398.65	USD 1398.65	USD 1398.65	USD 1398.65	USD 1398.65	USD 1398.65	USD 1398.65
**Transportation (local)**	USD 561.30	USD 561.30	USD 561.30	USD 561.30	USD 561.30	USD 561.30	USD 561.30
**Overheads**	USD 36.81	USD 36.81	USD 36.81	USD 36.81	USD 36.81	USD 36.81	USD 36.81
**Follow-Up Care**							
**Post-Treatment Consultations**	USD 8244.69	USD 8244.69	USD 8244.69	USD 8244.69	USD 8244.69	USD 8244.69	USD 8244.69
**Computed Tomography (CT Scan with Contrast)**	USD 5594.61	USD 5594.61	USD 5594.61	USD 5594.61	USD 5594.61	USD 5594.61	USD 5594.61
**Complete Blood Count with Differential**	USD 2149.51	USD 2149.51	USD 2149.51	USD 2149.51	USD 2149.51	USD 2149.51	USD 2149.51
**Chemistry Panel 7**	USD 265.01	USD 265.01	USD 265.01	USD 265.01	USD 265.01	USD 265.01	USD 265.01
**Renal Panel 1**	USD 1148.37	USD 1148.37	USD 1148.37	USD 1148.37	USD 1148.37	USD 1148.37	USD 1148.37
**Liver Function Tests**	USD 1001.14	USD 1001.14	USD 1001.14	USD 1001.14	USD 1001.14	USD 1001.14	USD 1001.14
**Cytology Smear**	USD 441.68	USD 441.68	USD 441.68	USD 441.68	USD 441.68	USD 441.68	USD 441.68
**Glycated Haemoglobin (HbA1c)**	USD 265.01	USD 265.01	USD 265.01	USD 265.01	USD 265.01	USD 265.01	USD 265.01
**Total Cholesterol**	USD 471.13	USD 471.13	USD 471.13	USD 471.13	USD 471.13	USD 471.13	USD 471.13
**Total (Unadjusted)**	**USD 107,019.68**	**USD 57,495.42**	**USD 64,764.74**	**USD 67,784.00**	**USD 88,458.85**	**USD 99,785.69**	**USD 86,811.35**
**Total (Adjusted)**	**USD 115,822.09**	**USD 60,434.07**	**USD 73,567.15**	**USD 76,586.41**	**USD 97,261.27**	**USD 108,588.11**	**USD 89,750.00**

Unadjusted: the costs of healthcare imports are excluded from the analysis; adjusted: the costs of healthcare imports are included in the analysis; ^m^ healthcare imports (cancer related services and/or associated costs linked to overseas care—PET Scan, Brachytherapy); ** basic systemic therapy consisting of Cisplatin + 5 Fluorouracil; *** targeted therapy with optional drugs for disseminated disease; a refers to surgery as an optional treatment for stage 1 disease; b refers to radiation therapy as an optional treatment for stage 1 disease; c refers to radiation therapy plus systemic therapy as an optional treatment for stage 1 disease.

**Table 4 ijerph-21-01685-t004:** Total annual cost estimation for cervical cancer (direct medical costs)—with preferred treatment of surgery for early-stage disease.

Parameter	Care Component/Procedures	Estimated Number of Cases in a Single year (N = 5)	Estimated Direct Medical Unit 2021 (USD)	Total Annual Direct Medical Costs (USD)	Percentage of Total Costs (%)	Range (USD) ±25%	
					(Adjusted)	Lower (−25%)	Upper (+25%)
**Diagnostic Process**	**Diagnostic Process**						
	Consultation (Clinical Assessment, Physical Examination)	5	USD 257.65	USD 1288.25		USD 966.19	USD 1610.31
	Cytology (Pap) Smear	3	USD 55.21	USD 165.63		USD 124.22	USD 207.04
	Colposcopy	5	USD 1473.74	USD 7368.70		USD 5526.53	USD 9210.88
	Biopsy	5	USD 1583.98	USD 7919.90		USD 5939.93	USD 9899.88
	Complete Blood Workup	5	USD 408.55	USD 2042.75		USD 1532.06	USD 2553.44
	Cytopathology	5	USD 478.49	USD 2392.45		USD 1794.34	USD 2990.56
	Imaging Studies	5	USD 1012.18	USD 5060.90		USD 3795.68	USD 6326.13
** *Subtotal* **				** *USD 26,238.58* **	** *11.00%* **	** *USD 19,678.94* **	** *USD 32,798.23* **
**Treatment**	**Treatment *(est. 3 cases for brachytherapy value = USD 17,591.28)***						
	Stage I-1b/IIa	1	USD 5705.03	USD 5705.03		USD 4278.77	USD 7131.29
	Stage IIb-II	1	USD 21,857.36	USD 21,857.36		USD 16,393.02	USD 27,321.70
	Stage III	2	USD 34,713.16	USD 69,426.32		USD 52,069.74	USD 86,782.90
	Stage IV	1	USD 15,875.05	USD 15,875.05		USD 11,906.29	USD 19,843.81
** *Subtotal* **				** *USD 112,863.76* **	** *47.33%* **	** *USD 84,647.82* **	** *USD 141,079.70* **
**Post-Treatment Side-effects Care**	**Post-Treatment Side-Effect-Relieving Procedures**						
	Urinary Retention	1	USD 1557.66	USD 1557.66		USD 1168.25	USD 1947.08
	Urinary Incontinence	1	USD 3516.18	USD 3516.18		USD 2637.14	USD 4395.23
	Renal Complications/Renal Failure	1	USD 20,674.86	USD 20,674.86		USD 15,506.15	USD 25,843.58
	Anaemia (Low Haemoglobin/Haematocrit)	3	USD 2461.54	USD 7384.62		USD 5538.47	USD 9230.78
	Urinary Tract Infections	5	USD 106.89	USD 534.45		USD 400.84	USD 668.06
	Other Complications of Treatment (post-operative pain, bowel issues, etc.)	5	USD 6747.76	USD 33,738.80		USD 25,304.10	USD 42,173.50
** *Subtotal* **				** *USD 67,406.57* **	** *28.27%* **	** *USD 50,554.93* **	** *USD 84,258.21* **
**Other Direct Costs**	**Other Direct Costs**						
	Nutrition Counselling	5	USD 100.00	USD 500.00		USD 375.00	USD 625.00
	Psychiatric/Psychological Counselling	5	USD 128.82	USD 644.10		USD 483.08	USD 805.13
	Pharmacy Services	5	USD 59.99	USD 299.95		USD 224.96	USD 374.94
	Imaging (PET scan—overseas) **	5	USD 1540.00	USD 7700.00		USD 5775.00	USD 9625.00
	Emergency Kit	5	USD 112.94	USD 564.70		USD 423.53	USD 705.88
	Transportation/Accommodation (imaging/treatment—overseas) **	5	USD 1398.65	USD 6993.25		USD 5244.94	USD 8741.56
	Transportation (local services related)	5	USD 561.30	USD 2806.50		USD 2104.88	USD 3508.13
	Overheads	5	USD 36.81	USD 184.05		USD 138.04	USD 230.06
** *Subtotal* **				** *USD 19,692.55* **	** *8.26%* **	**USD 27,962.87**	**USD 46,604.79**
**Follow-Up Care**	**Follow-Up Care**						
	Post-Treatment Consultations	5	USD 1030.59	USD 5152.95		USD 3864.71	USD 6441.19
	Computed Tomography (CT scan with contrast)	5	USD 699.33	USD 3496.65		USD 2622.49	USD 4370.81
	Complete Blood Count with Differential	5	USD 268.69	USD 1343.45		USD 1007.59	USD 1679.31
	Chemistry Panel 7	5	USD 33.13	USD 165.65		USD 124.24	USD 207.06
	Renal Panel 1	5	USD 143.55	USD 717.75		USD 538.31	USD 897.19
	Liver Function Tests	5	USD 125.14	USD 625.70		USD 469.28	USD 782.13
	Cytology Smear	5	USD 55.21	USD 276.05		USD 207.04	USD 345.06
	Glycated Haemoglobin (HbA1c)	5	USD 33.13	USD 165.65		USD 124.24	USD 207.06
	Total Cholesterol	5	USD 58.89	USD 294.45		USD 220.84	USD 368.06
** *Subtotal* **				** *USD 12,238.30* **	** *5.13%* **	** *USD 9178.73* **	** *USD 15,297.88* **
**Total Direct Medical Costs (unadjusted)**				**USD 206,155.23**		**USD 154,616.42**	**USD 257,694.04**
**Total Direct Medical Costs (adjusted)**				**USD 238,439.76**		**USD 178,829.82**	**USD 298,049.70**
**Total Direct Medical Costs (adjusted)-b**				**USD 251,572.84**		**USD 188,679.63**	**USD 314,466.05**
**Total Direct Medical Costs (adjusted)-c**				**USD 254,592.10**		**USD 190,944.08**	**USD 318,240.13**
**Total Direct Medical Costs (adjusted)-d**				**USD 182,007.88**		**USD 136,505.91**	**USD 227,509.85**
**Total Direct Medical Costs (adjusted)-e**				**USD 357,659.64**		**USD 268,244.73**	**USD 447,074.55**

** Healthcare imports (cancer related services plus associated costs linked to overseas care, referring to brachytherapy and PET scans); unadjusted: the costs of healthcare imports are excluded from the analysis; adjusted: the costs of healthcare imports are included in the analysis; b: adjusted total direct medical costs, reflecting radiation therapy as an optional treatment for early-stage disease; c: adjusted total direct medical costs, reflecting radiation therapy and chemotherapy as an optional treatment for early-stage disease; d: adjusted total direct medical costs, reflecting a 50% reduction in the direct medical unit cost of treatment for each cancer stage (preferred surgery for early-stage disease retained); e: adjusted total direct medical costs, reflecting a 50% increase in the direct medical unit costs of all care components (model with preferred surgery as treatment option for early-stage disease).

**Table 5 ijerph-21-01685-t005:** Total annual cost estimation for cervical cancer (direct medical costs) (with preferred treatment of surgery for early-stage disease).

Parameter	Care Component/Procedures	Estimated Number of Cases in a Single Year (N = 8)	Estimated Direct Medical Unit 2021 (USD)	Total Annual Direct Medical Costs (USD)	Percentage of Total Costs (%)	Range (USD) ±25%	
					(Adjusted)	Lower (−25%)	Upper (+25%)
**Diagnostic Process**	**Diagnostic Process**						
	Consultation (Clinical Assessment, Physical Examination)	8	USD 257.65	USD 2061.20		USD 1545.90	USD 2576.50
	Cytology (Pap) Smear	3	USD 55.21	USD 165.63		USD 124.22	USD 207.04
	Colposcopy	8	USD 1473.74	USD 11,789.92		USD 8842.44	USD 14,737.40
	Biopsy	8	USD 1583.98	USD 12,671.84		USD 9503.88	USD 15,839.80
	Complete Blood Workup	8	USD 408.55	USD 3268.40		USD 2451.30	USD 4085.50
	Cytopathology	8	USD 478.49	USD 3827.92		USD 2870.94	USD 4784.90
	Imaging Studies	8	USD 1012.18	USD 8097.44		USD 6073.08	USD 10,121.80
** *Subtotal* **				** *USD 41,882.35* **	** *11.44%* **	** *USD 31,411.76* **	** *USD 52,352.94* **
**Treatment**	**Treatment *(est. 5 cases for brachytherapy value = USD 29,318.80)***						
	Stage I-1b/IIa	1	USD 5705.03	USD 5705.03		USD 4278.77	USD 7131.29
	Stage IIb-II	2	USD 21,857.36	USD 43,714.72		USD 32,786.04	USD 54,643.40
	Stage III	3	USD 34,713.16	USD 104,139.48		USD 78,104.61	USD 130,174.35
	Stage IV	2	USD 15,875.05	USD 31,750.10		USD 23,812.58	USD 39,687.63
** *Subtotal* **				** *USD 185,309.33* **	** *50.60%* **	** *USD 138,982.00* **	** *USD 231,636.66* **
**Post-Treatment Side-Effects Care**	**Post-Treatment Side-Effect-Relieving procedures**						
	Urinary retention	1	USD 1557.66	USD 1557.66		USD 1168.25	USD 1947.08
	Urinary incontinence	1	USD 3516.18	USD 3516.18		USD 2637.14	USD 4395.23
	Renal Complications/Renal Failure	1	USD 20,674.86	USD 20,674.86		USD 15,506.15	USD 25,843.58
	Anaemia (Low Haemoglobin/Haematocrit)	3	USD 2461.54	USD 7384.62		USD 5538.47	USD 9230.78
	Urinary Tract Infections	8	USD 106.89	USD 855.12		USD 641.34	USD 1068.90
	Other Complications of Treatment (post-operative pain, bowel issues, etc.)	8	USD 6747.76	USD 53,982.08		USD 40,486.56	USD 67,477.60
** *Subtotal* **				** *USD 87,970.52* **	** *24.02%* **	** *USD 65,977.89* **	** *USD 109,963.15* **
**Other Direct Costs**	**Other direct costs**						
	Nutrition Counselling	8	USD 100.00	USD 800.00		USD 600.00	USD 1000.00
	Psychiatric/Psychological Counselling	8	USD 128.82	USD 1030.56		USD 772.92	USD 1288.20
	Pharmacy Services	8	USD 59.99	USD 479.92		USD 359.94	USD 599.90
	Imaging (PET scan—overseas) **	8	USD 1540.00	USD 12,320.00		USD 9240.00	USD 15,400.00
	Emergency Kit	8	USD 112.94	USD 903.52		USD 677.64	USD 1129.40
	Transportation/Accommodation (imaging/treatment—overseas) **	8	USD 1398.65	USD 11,189.20		USD 8391.90	USD 13,986.50
	Transportation (local services related)	8	USD 561.30	USD 4490.40		USD 3367.80	USD 5613.00
	Overheads	8	USD 36.81	USD 294.48		USD 220.86	USD 368.10
** *Subtotal* **				** *USD 31,508.08* **	** *8.60%* **	** *USD 23,631.06* **	** *USD 39,385.10* **
**Follow-up Care**	**Follow-up Care**						
	Post-Treatment Consultations	8	USD 1030.59	USD 8244.72		USD 6183.54	USD 10,305.90
	Computed Tomography (CT scan with contrast)	8	USD 699.33	USD 5594.64		USD 4195.98	USD 6993.30
	Complete Blood Count with Differential	8	USD 268.69	USD 2149.52		USD 1612.14	USD 2686.90
	Chemistry Panel 7	8	USD 33.13	USD 265.04		USD 198.78	USD 331.30
	Renal Panel 1	8	USD 143.55	USD 1148.40		USD 861.30	USD 1435.50
	Liver Function Tests	8	USD 125.14	USD 1001.12		USD 750.84	USD 1251.40
	Cytology Smear	8	USD 55.21	USD 441.68		USD 331.26	USD 552.10
	Glycated Haemoglobin (HbA1c)	8	USD 33.13	USD 265.04		USD 198.78	USD 331.30
	Total Cholesterol	8	USD 58.89	USD 471.12		USD 353.34	USD 588.90
** *Subtotal* **				** *USD 19,581.28* **	** *5.35%* **	** *USD 14,685.96* **	** *USD 24,476.60* **
**Total Direct Medical Costs (unadjusted)**				**USD 313,423.56**		**USD 235,067.67**	**USD 391,779.45**
**Total Direct Medical Costs (adjusted)**				**USD 366,251.56**		**USD 274,688.67**	**USD 457,814.45**
**Total Direct Medical Costs (adjusted)-b**				**USD 379,107.35**		**USD 284,330.51**	**USD 473,884.19**
**Total Direct Medical Costs (adjusted)-c**				**USD 382,403.90**		**USD 286,802.93**	**USD 478,004.88**
**Total Direct Medical Costs (adjusted)-d**				**USD 273,596.90**		**USD 205,197.67**	**USD 341,996.12**
**Total Direct Medical Costs (adjusted)-e**				**USD 549,377.34**		**USD 412,033.01**	**USD 686,721.68**

** Healthcare imports (cancer related services plus associated costs linked to overseas care); unadjusted: the costs of healthcare imports are excluded from the analysis; adjusted: the costs of healthcare imports are included in the analysis; b: adjusted total direct medical costs, reflecting radiation therapy as an optional treatment for early-stage disease; c: adjusted total direct medical costs, reflecting radiation therapy and chemotherapy as an optional treatment for early-stage disease; d: adjusted total direct medical costs, reflecting a 50% reduction in the direct medical unit cost of treatment for each cancer stage (preferred surgery for early-stage disease retained); e: adjusted total direct medical costs, reflecting a 50% increase in the direct medical unit costs of all care components (model with preferred surgery as treatment option for early-stage disease.

## Data Availability

All data generated or analysed during this study are included in the article. Data are fully available without restrictions and inquiries can be directed to the corresponding author.

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
