# Peer review of "Cost Analysis Related to Diagnosis, Treatment and Management of Cervical Cancer in Antigua and Barbuda: A Prevalence-Based Cost-of-Illness Study"

_ijerph, 2024, doi:10.3390/ijerph21121685_

Round 1
Reviewer 1 Report
Comments and Suggestions for Authors
I consider that the study addresses a relevant topic and can be useful for evaluating and planning cervical cancer control actions in the country; however, the inclusion of screening costs as part of the costs associated with cancer treatment does not seem appropriate. A comparison of the total screening costs for the entire asymptomatic population with the treatment costs would be more suitable. Furthermore, the methods need to more clearly define the procedures considered for each cancer staging.
The keywords are important for ensuring that your article can be located in searches; therefore, I suggest using MeSH descriptors, replacing the term "cervical" with "Uterine Cervical Neoplasms," and reevaluating the others. For example, "Cost-of-illness" and "Economic burden" are alternative terms, while "Direct medical costs" is not a MeSH term.
The description of the screening method employed in the country could be more explicit. From Figure 1, it appears that the screening is opportunistic, commencing at age 21, with cytological examinations and HPV DNA testing conducted every three years.
Table 1 does not include the cost of the HPV DNA test within the screening costs; however, it seems that this test is used in conjunction with cytology, rather than solely for follow-up care.
It is unclear how the authors calculated the costs associated with screening and diagnostic investigations. The data on cancer patients and the prevalence of cancer were insufficient to perform these calculations. Is there an estimation of how many women should be screened, what the expected positivity rate is, and how many will require a biopsy following a colposcopy showing abnormal findings? It may be pertinent to consider that screening costs should not be included in the cancer cost analysis, as the objective of screening is to identify precursor lesions rather than stage 3 and 4 cancers. It could be insightful if the authors calculated the costs associated with cancers identified outside the screening program (likely advanced cases) and the costs related to the treatment of precancerous lesions and cancer after a screening test.
The high cost of early-stage treatment is an unexpected finding. Do the authors have any hypotheses regarding these results? Was the management of side effects assessed in this study? Did the authors compare side effect rates by cancer stage? Is it possible to ascertain whether there are issues with patient care for those undergoing surgery in the country? In the methods section, it is not clear whether data from cases were utilized to calculate the average procedures used to treat cases at each stage or if the authors assumed that all patients required radiotherapy, chemotherapy, and procedures related to side effects, which are more commonly noted in advanced stages. The costs associated with stage III treatment appear to have been estimated considering only the absence of surgical intervention.
Although the authors report findings consistent with those in the literature, the cited studies indicate lower costs for carcinoma in situ. In various studies, including those referenced, costs tend to increase with higher staging, except for stage IV cases. The higher total cost in this study may be associated with a greater number of cases at advanced stages, while individual costs for early stages remain comparatively lower. For further reading, I recommend the article "Toward a Framework to Assess the Financial and Economic Burden of Cervical Cancer in Low- and Middle-Income Countries: A Systematic Review."
Author Response
|
Response to Reviewer 1 Comments
|
||
|
1. Summary |
|
|
|
Thank you very much for taking the time to review this manuscript. We do express our appreciations to you for your comments and suggestions offered. It is our hope that the revised manuscript has addressed your concerns. We do look forward to hearing from you on this. Please find the detailed responses below and the corresponding revisions/corrections highlighted in track changes in the re-submitted files.
|
||
|
2. Point-by-point response to Comments and Suggestions for Authors
|
||
|
Comments 1: I consider that the study addresses a relevant topic and can be useful for evaluating and planning cervical cancer control actions in the country; however, the inclusion of screening costs as part of the costs associated with cancer treatment does not seem appropriate. |
||
|
Response 1: Thank you for pointing this out. The authors have taken note of this and after consulting with experts wish to point out that screening costs as stated reflects ‘screening’ as part of the cervical cancer diagnostic process as would be determined for instance after someone presents with symptoms. This is so stated understanding that during the period 2017-2021, Antigua and Barbuda lacked an active or systematic screening program for cervical cancer. This is reflected in the last CanScreen 5 report on Antigua and Barbuda. Screening as a definitive header has been incorporated into diagnosis and the headers or care parameters now reflect the below named three parameters: Diagnostic process, Treatment, Follow-up care
Reference: International Agency for Research on Cancer. CanScreen 5; Country fact sheet: Antigua and Barbuda, CERVICAL CANCER SCREENING PROGRAMME. World Heal Organ 2023. https://canscreen5.iarc.fr/?page=countryfactsheetcervix&q=ATG&rc= (accessed November 27, 2024).
Comments 2: A comparison of the total screening costs for the entire asymptomatic population with the treatment costs would be more suitable. Response 2: Thank you for pointing this out. The authors have taken note of this, and do agree with the point understanding its significance in enhancing work of this type, however, we do wish to point out that to do this at this stage would be to act contrary to the scope of our study, and by extension the approvals as was granted by the requisite ethics bodies, to include gatekeeper permission which granted access to data for persons with definitively diagnosed cervical cancer and nothing else. So as not to lose the importance of the suggestion, however, the authors have since pointed this out in the study limitations and do also posit it as a recommendation to be addressed in future studies.
Lines 435-440 Comments 3: Furthermore, the methods need to more clearly define the procedures considered for each cancer staging. Response 3: Thank you for pointing this out. The authors wish to point out that this has since been done by firstly editing the subsection’s heading to accurately reflect the time period (2017-2019) during which cancer cases were detected and managed and having those changes highlighted in track changes of the paper under
‘2.4 Diagnosis and management of cervical cancer in Antigua and Barbuda (2017-2021)
The text in this subsection was revised with expert guidance/consultations as to what obtained in Antigua and Barbuda during the 2017-2021 period:
“2.4 Diagnosis and management of cervical cancer in Antigua and Barbuda during 2017-2021
In Antigua and Barbuda, most diagnosis and management of cancer is primarily conducted at the country’s lone tertiary care hospital, the Sir Lester Bird Medical Centre. Cancer management including diagnosis and treatment of cancer of the cervix is guided by treatment guidelines broadly adapted from World Health Organizations’ (WHO) and National Comprehensive Cancer Network Clinical Practice Guidelines (NCCN Guidelines) [23–25]. These guidelines provide guidance on optimizing care and improving management outcomes in respect of the diagnostic process, staging and treatment of cancer patients [26]. Prior to September 2022, and during the period 2017 to 2021, Antigua and Barbuda public health system lacked an active and/or systematic screening program for cervical cancer [27][28]. During the 2017 to 2021 period, cervical cancer diagnostic process would have started with female patients either consulting a general practitioner, or a Gynaecologist directly based on presenting symptoms (such as spontaneous or contact bleeding, pelvic pain, abnormal vaginal discharge, dyspareunia) or opportunistically (for instance, if patient is sexually active or patient is ≥ 21years) [29]. At gynaecological consultation, females would undergo a routine examination involving (i) physical assessment, (ii) clinical assessment, and (iii) a cytology smear (pap smear). Detection is based on the initial findings of physical examination in conjunction with the results of the cervical smear (Pap smear), all of which may or may not suggest the presence of abnormal cells in the cervix [29]. That is to say that the cells may be (i) negative for intraepithelial lesion (NLM), (ii) classified as Atypical Squamous Cells of Undetermined Significance (ASCUS), (iii) Low-Grade Squamous Intraepithelial Lesion (LSIL), (iv) High-Grade Squamous Intraepithelial Lesion (HSIL), (v) abnormal glandular cell or (v) malignant. Should the initial findings indicate the presence abnormal cells, then a cervical punch biopsy is taken at colposcopy for histopathologic assessment, and to determine whether cervical cells are precancerous (cervical intraepithelial neoplasia, CIN I- slightly abnormal cells affects up to 1/3 of epithelium, CIN II-abnormal cells occupy up to 2/3 of epithelium, CIN III-abnormal cells affect the full thickness of epithelium) or cancerous lesions [29]. Depending on the severity of the lesion as seen on Pap smear and the age of the patient then a histopathologic diagnosis is sought through colposcopy and biopsy. If the biopsy is positive for CIN, then this is treated by a Gynaecologist based on classification and in accordance with the American College of Obstetricians and Gynecologists (ACOG) guidelines [31]. Where the lesions are cancerous, then the biopsied specimen is further assessed to provide a histopathological stage for the disease. Pathological staging relies on the tumor node metastasis classification system (TNM) of the American Joint Committee on Cancer Staging System [32], and the International Federation of Obstetrics and Gynecology (FIGO) classification system [33]. Under the AJCC system the specimen is assessed to provide details on the size and spread of the tumour (T-primary tumour), how many lymph nodes are involved (if lymph nodes are in the sample) (N-regional lymph nodes), and whether there is pathological evidence of metastasis at a distant site (M-distant metastasis) [32]. With evidence of a positive biopsy for cancerous lesions (squamous cell carcinoma), the disease is then referred to a Medical Oncologist for clinical staging and management. Clinical staging entails the concomitant utilization of the results of the physical examination with colposcopy, and histopathology (cervical punch biopsy or conization), and is based on the International Federation of Obstetrics and Gynecology (FIGO) classification system [33]. This involves the use of results from (i) a complete blood workup; (ii) radiography (plain chest X-ray, abdominal/pelvic computed tomography scan with contrast, magnetic resonance imaging (MRI), electrocardiography), and (iii) endoscopy (cystoscopy or sigmoidoscopy if bladder or colon invasion is suspected). Clinical staging occurs before any treatment and is used to determine whether the disease is localized (stage 1-disease is confined to the cervix uteri; stage 2-disease extends beyond the uterus but does not affect the lower 1/3 of the vagina or pelvic wall) or disseminated (stage 3-disease is in lower 1/3 of the vagina and may extend to the pelvic wall and beyond; stage 4-extends into the bladder or rectum or organs or bones) as determined by the tumour size, depth of invasion and disease extent [34]. or disseminated based on the tumour size, depth of invasion and disease extent. Management is stage and disease extent dependent and may include surgical resection (hysterectomy usually for older females), concomitant radiation (external beam radiation therapy/brachytherapy -done overseas) and chemotherapy, or a combination of all [29] (Figure 1). Patients with localized disease (stages 1, 2) are offered (i) surgery followed by external beam radiation therapy and systemic therapy (chemotherapy) in that order, with patients needing to travel overseas for brachytherapy or (ii) surgery followed by chemoradiation (chemotherapy and external beam radiation given locally plus brachytherapy overseas). In some cases, post-operative patients may directly travel overseas for chemoradiation (chemotherapy plus external beam radiation and brachytherapy). Treatment of advanced or disseminated cervical cancer (stages 2b, 3, 4a) involves the use of systemic (chemotherapy) in addition to external beam radiation therapy and brachytherapy. Treatment of disseminated disease, stage 4b, is solely palliative, with patients being offered systemic therapy with or without external beam radiation. Further, given peculiarities that may exist for presentation patients, if a person presents with signs and symptoms of higher stage disease then screening would not be done. A tissue diagnosis is usually obtained, clinical diagnosis made and then the patient would commence treatment. Fertility sparing surgery, brachytherapy and positron emission tomography (PET scan) are currently not available in Antigua and Barbuda [34]. Cervical cancer patients requiring these services are referred to other countries for treatment. Except for fertility-preserving surgery, the Medical Benefits Scheme covers all cervical cancer-related costs for treatment abroad. In this study, cost estimates for all care components involved in the diagnosis and management of cervical cancer were based on 2021 market prices as per sources expressed in terms of private clinics and laboratories and tertiary hospital unsubsidized prices (Table 1).”
Comments 4: The keywords are important for ensuring that your article can be located in searches; therefore, I suggest using MeSH descriptors, replacing the term "cervical" with "Uterine Cervical Neoplasms," and reevaluating the others. For example, "Cost-of-illness" and "Economic burden" are alternative terms, while "Direct medical costs" is not a MeSH term. Response 4: Thank you for pointing this out. The authors wish to point out that the following words have been identified as keywords based on a rechecking of MESH terms : “Keywords: Antigua and Barbuda, Uterine cervical neoplasms, Cervical cancer, Cost of illness, Economic burden, cost analysis”
Comments 5: The description of the screening method employed in the country could be more explicit. From Figure 1, it appears that the screening is opportunistic, commencing at age 21, with cytological examinations and HPV DNA testing conducted every three years. Response 5: Thank you for pointing this out. The authors have since edited the information on screening to only indicate that cytological examination was done every three years (Lines . Further the authors wish to share that HPV DNA testing was not included in text surrounding opportunistic screening as it was not a feature of the diagnostic algorithm used Antigua and Barbuda during the period 2017-2021. We thus refrained from including the cost of HPV DNA test within the screening costs so as to ensure that our cost estimates match the scope of our investigation, which covers a period prior to the implementation of HPV testing in Antigua and Barbuda (HPV testing was introduced in September 2022, via a pilot program). The authors do however noted its absence at this time to be a limitation and has recommended ‘in our discussion/conclusion’ that future studies to look at the cost of cervical cancer, post implementation of HPV testing and local screening guidelines would be beneficial.
References: The Daily Observer. Pioneering pilot project for cervical cancer screening being launched in Antigua and Barbuda today. 2022 September 7. Dly Obs Newsp 2022. https://antiguaobserver.com/pioneering-pilot-project-for-cervical-cancer-screening-being-launched-in-antigua-and-barbuda-today/ (accessed November 27, 2024).
The Ministry of Health, Wellness ST and the E of A and B. ANTIGUA AND BARBUDA NATIONAL GUIDELINES FOR CERVICAL SCREENING & TREATMENT OF PRE-CANCER LESION. Gov Antig Barbuda 2024. https://health.gov.ag/antigua-and-barbuda-national-guidelines-forcervical-screening-treatment-ofpre-cancer-lesions-quick-reference-guid/ (accessed November 28, 2024).
Comments 6: It is unclear how the authors calculated the costs associated with screening and diagnostic investigations. Response 6: Thanks for highlighting this. The authors wish to share that the costs associated with screening and diagnostic investigations were obtained from reimbursement receipts archived at the Medical Benefits Scheme as well as per cost information obtained from private health facilities. This information was then used in to compute costs based on the notes given in the text re: subsection ‘2.4 Cost data’
Direct medical costs were computed by multiplying the number of cases requiring healthcare by the corresponding costs of each resource per case. Total costs represented an aggregate of the costs of all care parameters. Direct Medical Costs of Disease (mc) = ∑(Mi x Pi) Mi is number of cases requiring health care Pi is the required health care resources unit costs per case mc is the total costs Costs were reported in 2021 USD. This was attained following adjustment of the country’s consumer price index (CPI) of 2021 and the USD 2021exchange rate (1 USD = 2.7169 XCD) as below: Value in 2021 USD=base year price × CPI in 2021/CPI in based year CPI in 2021 = 95.27; CPI in based year = 95.27
For example: The cost of a cytology smear in 2021 based on receipts seen was XCD 150.00 Converted to USD: 150.00 / 2.7169 = USD 55.21 the base year price Value in 2021 USD = 55.21 x (CPI in 2021/ CPI in base year) This gives Value in 2021 USD = 55.21 x (95.27 / 95.27) = USD 55.21
Therefore, for ease of understanding if an estimated 2 persons, diagnosed with early-stage cancer had cytology in a year, this would in effect be used to give the direct medical costs for cytology in a single year as: 2 x 55.21 = USD 110.42
Comments 6:The data on cancer patients and the prevalence of cancer were insufficient to perform these calculations. Response 6: The authors having taken keen note of the reviewer’s comment do offer that we differ strongly with this assertion. Our view is founded on the following: 1. Considering the country’s population of females (roughly 53% of population) and with 40 diagnosed cases of cervical cancer in the five-year period 2017-2021, this would amount roughly to an age-standardized rate of around 12 per 100,000 female population. And 2. Calculation of an estimate of cervical cancer cases in a single year The authors wish to point out that since the study is a prevalence-based cost-of-illness study, then by demonstrating how prevalence cases are accounted for in the methodology is important in helping to define, for reasons of cost analysis, the number of cases requiring healthcare.
In this regard, the authors have inserted under subsection 2.5 the below information so as to lend clarity or definition to the number of cases considered for the analysis.
For our cost analysis, we estimated the number of cervical cancer cases requiring healthcare in a single year in Antigua and Barbuda, computed by subtracting the number of deaths from the total number of cases and divided the answer by 5 [6]. For comparative purposes and to assess directly the impact of unrecorded cases at our study sites, the average number of cancer cases in a single year was increased by 50% and the total annual costs assessed and reported [6,27].
The authors posit that a similar approach was used by Ginindza and colleagues (Section Materials & Methods, subsection Cervical cancer, paragraph 3; Ref #6 on our list:
Ginindza TG, Sartorius B, Dlamini X, Östensson E. Cost analysis of Human Papillomavirus-related cervical diseases and genital warts in Swaziland. PLoS One 2017;12:e0177762. https://doi.org/10.1371/journal.pone.0177762.
Moreover, we have also included in the ‘Results’ section the following “3.2 Estimate of Cervical Cancer Cases in a Single Year
Information obtained from our study sites indicated that 14 of the 40 diagnosed cancer cases had died during the study period. We therefore estimated that there were 5 cases of cancer on average, in a single year in Antigua and Barbuda. This was calculated by subtracting the 14 deaths from the 40 cases diagnosed in the period and dividing the result (26) by 5. When the average number of cases in a single year (5) increased by 50%, the estimated number of cases per year changed to 8.
Comments 7: Is there an estimation of how many women should be screened, what the expected positivity rate is, and how many will require a biopsy following a colposcopy showing abnormal findings? Response 7: The authors have taken a keen note of the reviewer’s comments, and though we are inclined to agree with the observation, we wish to share that neither the scope of our study/work, study duration nor ethical approvals permitted us to gather data on the estimated number of women to be screened, positivity rate, and how many will require biopsy following colposcopy showing abnormal findings. It must be noted that before September 2022, with the launch of the Cervical Cancer Elimination program, this sort of information was not readily available. The authors further acknowledge this as a limitation and have highlighted same in the discussion/conclusion section of the manuscript. We also recommended that it would be beneficial for future studies to give attention to the relationship between or impact of screening and positivity rate for cervical cancer. Lines 435-440 Comments 8: It may be pertinent to consider that screening costs should not be included in the cancer cost analysis, as the objective of screening is to identify precursor lesions rather than stage 3 and 4 cancers. Response 8: The authors have noted the reviewer’s comments and though in agreement wish to indicate that only costs for ‘screening’ as it relates to the overall diagnostic process is considered (from the presentation of symptoms, to histopathology, to clinical staging). Having reviewed our notes in consultation with experts, and based on the reviewers comments we wish to point out that invariably if persons present with signs and symptoms of higher stage disease then screening would not be done. A tissue diagnosis is usually obtained, clinical diagnosis made and then the patient would commence treatment. The authors have since inserted a note to this effect in the text of subsection ‘2.4 Diagnosis and Management of Cervical Cancer in Antigua and Barbuda During 2017-2021’ The authors do thank the reviewer for urging this response.
Comments 9: It could be insightful if the authors calculated the costs associated with cancers identified outside the screening program (likely advanced cases) and the costs related to the treatment of precancerous lesions and cancer after a screening test. Response 9: The authors do wish to thank the reviewer for pointing out this observation and for making an insightful and useful recommendation. Notwithstanding this however, the authors wish to share that considering the ethical implications involved re: ethics approvals, study duration, scope of the study, retrospective design, and/or study objective, we can only at this time identify our inability to assess or evaluate costs related to treatment of precancerous lesions as a study limitation and recommend that this be considered in future cost studies and/or cost models. Our study population focused on cases who already had a definitive diagnosis of cervical cancer.
Comments 10: The high cost of early-stage treatment is an unexpected finding. Do the authors have any hypotheses regarding these results? Response 10: The authors do wish to thank the reviewer for this searching question. The authors have taken note of this with expert guidance, had cause to revisit, review and/or revise the included care components listed for each cancer stage. Having done so, we have had to revise downward the cost of early-stage treatment. This change is now reflected in our tables expressing our cost estimates. The authors do thank the reviewer for urging us to review our costs thus ensuring that they clearly reflects what obtains re: local treatment of cervical cancer by stage.
Comments 11: Was the management of side effects assessed in this study? Response 11: The authors wish to share that this information was considered based on expert input and is reflected under the care parameter ‘Post-treatment side-effects relieving procedures.’ The authors note that this list is not exhaustive but captures a key list of side-effects that frequently affect most of our cervical cancer patients (post-treatment) and for which management is required. Comments 11: Did the authors compare side effect rates by cancer stage? Response 11: Even though this was done prior, the authors nonetheless have reviewed this information with expert guidance, especially regarding each cancer stage and have made the appropriate adjustments in the tables. The authors wish to thank the reviewer for urging this response.
Comments 12: Is it possible to ascertain whether there are issues with patient care for those undergoing surgery in the country? Response 12: The authors noted the reviewer’s question and wish to share that re: surgery, during the 2017-2021 period and up to present date, patients would not have had any excessive waiting time for surgery since they had access to a surgical oncologist out of Trinidad and Tobago, who travels to Antigua and Barbuda every 6-8 weeks. Patients do not have to pay for these surgeries as they are financed by the government funded Medical Benefits Scheme. The country boasts in excess of twelve competent Obstetrician/ Gynaecologists (OB/GYN) who provide follow-up/post-surgical care with inputs from the surgical oncologist via long distance should any complications arise. Additionally, post-surgery patients have access to two medical oncologists, and excellent pharmaceutical care via the hospital and through the Medical Benefits Scheme. During the 2017-2021 period up until April 2023 they had access to EBRT at The Cancer Centre Eastern Caribbean and/or travel overseas with support from the Medical Benefits Scheme for chemoradiation or brachytherapy based on recommendations of the medical oncologist/multidisciplinary team. The authors having said this have ensured that points to this effect are noted in the manuscript re: section 2.4 Diagnosis and management of cervical cancer in Antigua and Barbuda (2017-2021) noted these points in the discussion section of the manuscript.
Comments 13: In the methods section, it is not clear whether data from cases were utilized to calculate the average procedures used to treat cases at each stage or if the authors assumed that all patients required radiotherapy, chemotherapy, and procedures related to side effects, which are more commonly noted in advanced stages. Response 13: The authors wish to thank the reviewer for pointing this out. The authors did had to use data from cases in conjunction with expert guidance to identify the key procedures used locally to treat cases at each stage of cervical cancer. In expressing our estimates of cost, it must be noted that all patients have access to these services as part of the general care algorithm in Antigua and Barbuda. Agreeably some procedures are more likely to feature at certain stages more than others. With further guidance the authors adjusted the key procedures used to treat cases at each stage by focusing on the primary or preferred treatment option per stage. Following this important adjustment, we can now say that the costs are now in alignment with the expected preferred treatment per stage.
Comments 14: The costs associated with stage III treatment appear to have been estimated considering only the absence of surgical intervention. Response 14: The authors wish to thank the reviewer for pointing this out. We have since reviewed this information and had cause to reach out again to the local medical oncologist for reconfirmation of the local approach (protocol) used in treating each stage of cervical cancer. We then reviewed our cost data obtained from archived reimbursement receipts and do wish to share that the cost information for each care component linked with each stage of cancer except for an adjustment made to correct an inadvertent overstatement of the cost of targeted therapy, all other costs are correctly stated.
Comments 15: Although the authors report findings consistent with those in the literature, the cited studies indicate lower costs for carcinoma in situ. In various studies, including those referenced, costs tend to increase with higher staging, except for stage IV cases. The higher total cost in this study may be associated with a greater number of cases at advanced stages, while individual costs for early stages remain comparatively lower. For further reading, I recommend the article "Toward a Framework to Assess the Financial and Economic Burden of Cervical Cancer in Low- and Middle-Income Countries: A Systematic Review." Response 15: The authors have taken keen note of the reviewer’s comments and was compelled to review all cost data, especially the data re: drugs and drugs combinations used across both systemic therapy and targeted therapy (formerly optional systemic therapy). While we did not had to make any adjustments to other components, for systemic therapy, it was necessary that we adjust the cost related to targeted therapy as it was inadvertently overstated by USD 643.83. The revised value for targeted therapy is now USD 12,855.79. Further and following expert guidance re: an earlier observation by the reviewer (comment 13), the authors had to make changes to allow for each stage to be costed correctly based on the primary or preferred treatment noted for that stage. Following these changes, we can confidently share that the cost estimates per stage of disease were found to be consistent with references which suggest that costs tend to increase with higher stage except for stage 4 disease. This in now reflected in both our results and discussion sections of the manuscript. The authors have also considered the recommended reference valuable to our research and have since cited it in our manuscript.
|
||
|
|
||
|
Kindly note that in addition to the edits done in respect of the comments and/or suggestions of the Reviewer, the authors have made some edits to further improve the article and so as to ensure that there is consistency across all areas of our study re: scope and/or purpose. This included edits to text, figures and tables. |
||
|
|
||
|
Thank you |
||

Reviewer 2 Report
Comments and Suggestions for Authors
REVIEW
This study looks at the medical costs of cervical cancer in Antigua and Barbuda, focusing on how much it costs to screen, diagnose, treat, and manage the disease. Here are key recommendations for author(s) to improve:
Introduction and Literature Review:
The introduction could benefit from more detailed comparisons with similar cost analyses from other low- and middle-income countries. This could enhance the contextual understanding of the study's findings. Include, if possible, more recent references, particularly those relevant to cervical cancer cost studies in regions with similar healthcare structures to Antigua and Barbuda.
Methodology:
Provide additional clarification on the rationale for selecting specific cost parameters, such as the choice of direct costs only. Explain why indirect costs were excluded and discuss how their inclusion might alter the findings. The study utilizes a prevalence-based cost-of-illness approach, but the justification for this method versus other potential cost-estimation methods could be better articulated.
Data Sources and Limitations:
Specify any potential limitations related to data collection from different health institutions and how these might affect the accuracy of cost estimates. Given that expert consultations contributed to data, the paper should address potential biases from these consultations and how they were minimized.
Discussion and Conclusion:
The discussion should better address the implications of these cost findings for healthcare policy and future research, particularly regarding cost containment and cervical cancer prevention strategies.
Formatting and Tables:
Some tables, especially Table 4 detailing cost estimates, could be simplified for readability. For instance, grouping similar cost categories could make it easier to follow. Ensure consistency in the use of terms like "adjusted" and "unadjusted" across the text and tables to avoid confusion.
Author Response
|
Response to Reviewer 2 Comments
|
||
|
|
|
|
|
Thank you very much for taking the time to review this manuscript. We do express our appreciations to you for your comments and suggestions offered. It is our hope that the revised manuscript has addressed your concerns. We do look forward to hearing from you on this. Please find the detailed responses below and the corresponding revisions/corrections highlighted in track changes in the re-submitted files.
|
||
|
2. Point-by-point response to Comments and Suggestions for Authors
|
||
|
Comments 1: Introduction and Literature Review: The introduction could benefit from more detailed comparisons with similar cost analyses from other low- and middle-income countries. This could enhance the contextual understanding of the study's findings. Include, if possible, more recent references, particularly those relevant to cervical cancer cost studies in regions with similar healthcare structures to Antigua and Barbuda. |
||
Response 1: The authors wish to thank the reviewer for this insightful and instructive comment. We have since incorporated this change in one of the paragraphs in the ‘introduction section' of the manuscript.
Lines 52-62.
“Introduction
The burden of cervical cancer continues to be a significant public health challenge worldwide [1]. Cervical cancer is one of the commonest cancers in terms of incidence and mortality among women [1]. In 2020, cervical cancer accounted for about 604,000 new cases and 342,000 cancer deaths, with its burden disproportionately more in low income countries compared to high income countries [1].
In Antigua and Barbuda, cervical cancer is ranked among the most common cancer among women in terms of incidence and mortality [2,3]. Cervical cancer, is known to have an infectious origin from the preventable human papilloma virus (HPV) [1]. Cervical cancer places a considerable burden on health systems given its direct and indirectly associated management costs (screening, diagnosis, management)[4]. Evidence shows that the costs associated with cervical cancer varies widely by countries [5]. Ginindza et al., 2017, estimated the total annual direct medical costs associated with screening, managing and treating cervical lesions, cervical cancer and genital warts in Eswatini formerly Swaziland at US$16 million [6]. In an earlier study done in Malaysia, the cost of managing cervical cancer in the public setting was USD 75,888,329.45 [7], while in India, Singh et al., 2020, reported the cost associated with cervical cancer as ranging from INR 19,494 to 41,388 (US$291 – 617) [8]. Further, in reporting on the economic burden of cervical cancer in Eswatini, Ngcamphalala and colleagues, estimated direct costs to be USD 13.7 million [9], a value similar to USD 12,589,360 estimated by Berraho et al., 2012 for Morocco. Moreover, the findings of a real-world study, showed that the aggregate direct medical costs of cervical cancer, which accounted for hospitalizations, appointments and procedures done across three Latin-American countries, namely, Brazil, Columbia and Mexico, varied considerably from USD 46, 514,395.93 in Brazil, to USD52,862,760.88 and USD 30,174,473.64, for Columbia and Mexico, respectively [10]. Thus, underscoring the vast disparity in the cost burden of cervical cancer across low to middle income countries.
Globally, the economic costs associated with cervical cancer is projected to be on the rise [1]. Information on the economic burden of cervical cancer in Antigua and Barbuda is lacking. This study, therefore, aimed to estimate the economic burden of cervical cancer in Antigua and Barbuda from the healthcare provider’s perspective.
Comments 2: Methodology:
Provide additional clarification on the rationale for selecting specific cost parameters, such as the choice of direct costs only.
Response 2: The authors have revised the title and edited the introduction as well as included a subsection in the methodology so as to ensure that there is no ambiguity re: the study scope, study objective, methodology and outcome. Further the inclusion of considerations for indirect costs have been noted in the discussion section.
Comments 3: The study utilizes a prevalence-based cost-of-illness approach, but the justification for this method versus other potential cost-estimation methods could be better articulated.
Response 3: The authors have taken note of this comment and have since inserted in the Method section under Costing the following:
“2.6 Cost-of-illness
This was a prevalence-based cost-of-illness (COI) study conducted from the healthcare provider’s perspective [37]. Use of the cost-of-illness method was premised on its general usefulness as a simple and easy tool for identifying the drivers of diagnosis and treatment costs associated with several diseases [38]. These studies type of studies may be described as being prevalence-based or incidence-based (77), with prevalence-based studies used to estimate the economic burden or total costs of a condition over a specified period (usually a year), and incidence-based studies used in estimating the lifetime costs of an illness from diagnosis until its disappearance or another endpoint (82,108,113).
For conditions that are profoundly burdensome on small countries such as those in the English-speaking Caribbean with their budgetary restrictions and relatively disadvantaged health system, cost studies using the cost-of-illness methodology have the potential to empower public health policymakers and healthcare providers with information on specific diseases or conditions [38, 39]. Usage of the COI methodology therefore is an enabler for the identification, quantification and valuation of resources related to any named condition from several perspectives, inclusive of international, governmental, societal and healthcare provider [39, 40]. This study therefore made ample use of the prevalence based cost-of-illness method in estimating economic burden of cervical cancer, a chronic disease that weighs heavily on health expenditure in Antigua and Barbuda [41].
Healthcare interventions required in cervical cancer management continuum (diagnosis and management) were identified (micro-costing approach), quantified and valued per case and extrapolated using prevalence data to estimate national costs. Data were collected retrospectively using patient charts and excel spreadsheet.
Direct Medical Costs of Disease (mc) = ∑(Mi x Pi)
Mi is number of cases requiring health care
Pi is the required health care resources unit costs per case
mc is the total costs
We investigated only direct medical costs given its usefulness in evaluating disease burden in the healthcare setting [42] [43] and determined same using a top-down and bottom-up approach [37]. Direct medical costs considered recurrent costs [44]. Recurrent costs included personnel, travel, consumables supplies (medical and non-medical), treatment, administration, and overheads [44].
For our cost analysis, we estimated the number of cervical cancer cases requiring healthcare in a single year in Antigua and Barbuda, computed by subtracting the number of deaths from the total number of cases and divided the answer by 5 [44].”
Comments 4: Data Sources and Limitations:
Specify any potential limitations related to data collection from different health institutions and how these might affect the accuracy of cost estimates. Given that expert consultations contributed to data, the paper should address potential biases from these consultations and how they were minimized.
Response 4: The authors have taken note of this and have since inserted a note in the discussion section to best capture our response to this query;
Re: biases how mitigated /sensitivity analysis to account for uncertainty in cost mode
Lines 447-450
Comments 5: Discussion and Conclusion:
The discussion should better address the implications of these cost findings for healthcare policy and future research, particularly regarding cost containment and cervical cancer prevention strategies.
Response 5: The authors have taken note of these points and have since made some changes to both the discussion and conclusion sections of the manuscript:
Comments 6: Formatting and Tables:
Some tables, especially Table 4 detailing cost estimates, could be simplified for readability. For instance, grouping similar cost categories could make it easier to follow. Ensure consistency in the use of terms like "adjusted" and "unadjusted" across the text and tables to avoid confusion.
Response 6: The authors have reviewed the tables in respect of the comments shared and do wish to indicate that one change was wrought where we replaced screening with the words “diagnostic process’ thereby combining ‘screening’ with ‘diagnosis.’ This was done to allow for our estimates to accurately reflect the procedures involved in the diagnosis and management of cervical cancer.
Note that the words adjusted and unadjusted were retained given their importance in defining inclusion and exclusion of the requisite healthcare imports.
|
Kindly note that in addition to the edits done in respect of the comments and/or suggestions of the Reviewer, the authors have made some edits to further improve the article and so as to ensure that there is consistency across all areas of our study re: scope and/or purpose. This included edits to text, figures and tables. |
|
|
|
Thank you |

Reviewer 3 Report
Comments and Suggestions for Authors
This is a well-written manuscript with clear and fluent language that is easy to understand. The study focuses on an important topic—evaluating the costs of cervical cancer in Antigua and Barbuda. The authors have applied a prevalence-based cost-of-illness methodology to estimate the direct medical costs of cervical cancer. The study's findings suggest that treatment is the primary cost driver and highlight the financial benefits of emphasizing prevention and early detection. Below are a few questions and points for consideration:
1. The manuscript clearly identifies the data source, with 40 patients found in the source sites. My first question is: how representative are these 40 patients of the broader population? Were there any other patients in Antigua and Barbuda who were not included in the study? If so, how many? Since the study employs a prevalence-based methodology, this question is important for assessing the generalizability of the results.
2. The manuscript mentions that, given the 14 patients who died during the study period, the authors estimated an average of 5 patients per year and excluded the associated costs of patients who died. However, the costs in the final phase of cancer are usually much higher than in earlier stages. Therefore, the costs for the patients who died during the study period may have been significantly higher than those of the surviving patients. It might be useful to reconsider the exclusion of these patients from the cost analysis.
3. Given that all patients follow different clinical pathways, experience varying complications, and receive different treatments, the estimated costs may not fully capture the real costs. The costs for individual patients can vary significantly. In Table 2, the authors describe the distribution of cancer stages, and in Table 3, they show the unit costs for different stages. Did the authors account for the cancer stage when calculating the total costs for each patient?
4. How was the sensitivity analysis conducted? What parameters were varied during the analysis, and how might these variations affect the results?
Author Response
Response to Reviewer 3 Comments
|
1.Thank you very much for taking the time to review this manuscript. We do express our appreciations to you for your comments and suggestions offered. It is our hope that the revised manuscript has addressed your concerns. We do look forward to hearing from you on this. Please find the detailed responses below and the corresponding revisions/corrections highlighted in track changes in the re-submitted files.
|
|
2. Point-by-point response to Comments and Suggestions for Authors
|
Comments 1: The manuscript clearly identifies the data source, with 40 patients found in the source sites. My first question is: how representative are these 40 patients of the broader population?
The information on the 40 patients were identified via record abstraction from key study sites, namely the Sir Lester Bird Medical Centre, Medical Benefits Scheme, The Cancer Centre Eastern Caribbean.
Response 1: The authors have identified this as a strength of the study results and do offer that having or advocating for a population- based cancer registry with accompanying legislative support to make reporting of cancer cases mandatory would be useful in future research.
Thus, the approach taken underscores the fact that there is no cancer registry on island, whether hospital-based or population based, hence the approach of record abstraction taken.
Notwithstanding the current challenges in obtaining data from private facilities, the authors wish to also point out that the key study sites combined accounts for the bulk of documented evidence on cancers in the country. Words to this effect have since been used to identify this observation in the discussion/conclusion section of the manuscript.
Additionally, the authors wish to point out that since the study is a prevalence-based cost-of-illness study, then by demonstrating how prevalence cases are accounted for in the methodology is important in helping to define, for reasons of cost analysis, the number of cases requiring healthcare.
In this regard, the authors have inserted under subsection 2.6 the below information so as to lend clarity or definition to the number of cases considered for the analysis.
“For our cost analysis, we estimated the number of cervical cancer cases requiring healthcare in a single year in Antigua and Barbuda, computed by subtracting the number of deaths from the total number of cases and divided the answer by 5 [37]. For comparative purposes, the average number of cases in a single year was increased by 50% and the total annual costs reassessed and reported [46]. The direct medical unit costs of care components were also increased by 50% and the total annual costs reassessed and reported [46]. Additionally, treatment cost in the original model was reduced by 50%, costs of other parameters remained unchanged and the impact of this manipulation on the total annual costs was assessed and reported.”
The authors also posit that a similar approach was used by Ginindza and colleagues (Section Materials & Methods, subsection Cervical cancer, paragraph 3; Ref #6 on our list:
Ginindza TG, Sartorius B, Dlamini X, Östensson E. Cost analysis of Human Papillomavirus-related cervical diseases and genital warts in Swaziland. PLoS One 2017;12:e0177762. https://doi.org/10.1371/journal.pone.0177762.
Moreover, we have also included in the ‘Results’ section the following
“3.2 Estimate of Cervical Cancer Cases in a Single Year
Information obtained from our study sites indicated that 14 of the 40 diagnosed cancer cases had died during the study period. We therefore estimated that there were 5 cases of cancer on average, in a single year in Antigua and Barbuda. This was calculated by subtracting the 14 deaths from the 40 cases diagnosed in the period and dividing the result (26) by 5.
When the average number of cases in a single year (5) increased by 50%, the estimated number of cases per year changed to 8.
Comments 2: Were there any other patients in Antigua and Barbuda who were not included in the study? If so, how many? Since the study employs a prevalence-based methodology, this question is important for assessing the generalizability of the results.
Response 2a: Given the absence of a population-based/hospital-based cancer registry and understanding that study data on cancer cases was only obtained from agencies for which the authors were granted gatekeeper permission and/or IRB approval, it is possible that not all cases of cervical cancer across the country would have been accounted for during the period 2017-2021.
Currently private facilities, including laboratories and clinicians in the private sector are not mandated to report on cancer cases diagnosed and/or treated in their facilities.
Notwithstanding this limitation, however, the authors posit that because our named study sites hold copious data on cancer cases, diagnosed, treated and/managed on the island by virtue of their roles, we consider the information obtained from these sites to be sufficiently reliable to enable generalizability of the results. This is so especially given that, aside from the absence of a cancer registry, cancer care in Antigua and Barbuda is mostly funded by the Medical Benefits Scheme.
Reference:
- Communications and Marketing Department Medical Benefits Scheme. About Us. Medical Benefits Scheme, https://www.mbs.gov.ag/v2/about/ (2023, accessed 22 September 2023).
Response 2b: The authors do understand the significant impact unrecorded cases could have on assessing the generalizability of the study estimates. To this end the authors have revised the subsection on ‘sensitivity analysis’ to show how unrecorded cases could best be accounted for in the cost estimation.
That is
“2.8 Sensitivity analysis
Using a method that was applied and mentioned in other studies, we performed sensitivity analysis using the lower and upper bound of ± 25% so as to account for the impact of uncertainty in the cost estimation and any unrecorded cases by the study sites used in our cost-of-illness study [13, 37, 42, 50].”
The authors posit that a similar approach was considered by:
Ngcamphalala C, Östensson E, Ginindza TG. The economic burden of cervical cancer in Eswatini: Societal perspective. PLoS One 2021; 16: e0250113.
Ginindza TG, Sartorius B, Dlamini X, Östensson E. Cost analysis of Human Papillomavirus-related cervical diseases and genital warts in Swaziland. PLoS One 2017;12:e0177762. https://doi.org/10.1371/journal.pone.0177762.
Comments 3: The manuscript mentions that, given the 14 patients who died during the study period, the authors estimated an average of 5 patients per year and excluded the associated costs of patients who died. However, the costs in the final phase of cancer are usually much higher than in earlier stages. Therefore, the costs for the patients who died during the study period may have been significantly higher than those of the surviving patients. It might be useful to reconsider the exclusion of these patients from the cost analysis.
Response 3: The authors have reviewed the comment and wish to share that in the absence of a population-based cancer registry and understanding that the information obtained from the Health Information Division, Ministry of Health clearly showed that there were 14 cervical cancer related deaths occurring for our cases for the period of data capture 2017-2021, then to not account for the deaths would make it difficult to compute a reliable estimate of the direct medical costs of cervical cancer, which should be based on an average of prevalent cases in a single year of the period of data capture.
In the absence of any other information that can affect prevalent cases, then accounting for the deaths, we consider, would make for the derivation of a more reliable estimate of study cost.
The authors have given thought to this and have decided to retain this information on the basis that this approach was considered in a similar study by Ginindza and colleagues:
Ginindza TG, Sartorius B, Dlamini X, Östensson E. Cost analysis of Human Papillomavirus-related cervical diseases and genital warts in Swaziland. PLoS One 2017;12:e0177762. https://doi.org/10.1371/journal.pone.0177762.
As indicated earlier, we have also included in the ‘Results’ section the following
“3.2 Estimate of Cervical Cancer Cases in a Single Year
Information obtained from our study sites indicated that 14 of the 40 diagnosed cancer cases had died during the study period. We therefore estimated that there were 5 cases of cancer on average, in a single year in Antigua and Barbuda. This was calculated by subtracting the 14 deaths from the 40 cases diagnosed in the period and dividing the result (26) by 5.
Further, understanding that there may be merit in retaining the 40 cases and thus allowing for comparisons to be made by having a much larger number of average prevalent cases, the authors, having since shared under subsection “sensitivity analysis” on the need to account for uncertainty due to possible unrecorded cases, and given that data capture was restricted to the four named study sites, have since included under the subsections ‘Cost-of-illness, ‘Estimate of cervical cancer cases in a single year’ and ‘Total annual direct medical costs’ the following:
“For comparative purposes, the average number of cancer cases in a single year was increased by 50% and the total annual costs reassessed and reported [46].”
And
When the average number of cases in a single year (5) increased by 50%, the estimated number of cases per year changed to 8.
And
Following the increase in the average number of prevalent cases by 50% (from 5 to 8 for a single year), the estimated total annual direct medical cost changed to USD 366,251.56 (ranged between USD 274,688.67 and USD 457,814.45), which represented a 54% increase in total annual direct medical costs (Table 5). This suggested that each additional prevalent case over and above 5 cases would be responsible for a USD 42,603.93 added to total annual cost.
(To demonstrate the effect of this change the authors have inserted a Table 5 and a Figure 3 into the manuscript)
The authors wish to share that a similar approach to what was done above was considered by Ginindza and colleagues:
(Section Results, subsection ‘Total annual estimated cost’, final paragraph:
Ginindza TG, Sartorius B, Dlamini X, Östensson E. Cost analysis of Human Papillomavirus-related cervical diseases and genital warts in Swaziland. PLoS One 2017;12:e0177762. https://doi.org/10.1371/journal.pone.0177762.
The variations in estimates observed when both average prevalent cases and direct medical unit cost per each care component were increased to 50% were given attention in the discussion section of the paper.
Comments 4: Given that all patients follow different clinical pathways, experience varying complications, and receive different treatments, the estimated costs may not fully capture the real costs.
The costs for individual patients can vary significantly. In Table 2, the authors describe the distribution of cancer stages, and in Table 3, they show the unit costs for different stages. Did the authors account for the cancer stage when calculating the total costs for each patient?
Response 4a: The authors are in agreement with the reviewer that costs for individual patients can vary significantly irrespective of whether they are commonalities in terms of similar stage and disease extent and also on the basis of whether all cases in the population were accounted for or not. It is on this very premise, therefore, that the authors have sought to include in the methodology a ‘sensitivity analysis.’
This was done to account for potential uncertainty in the model (likely variations in cost estimation per individual patients and any unrecorded cases). The authors relied on one-way sensitivity analysis, allowing for upper and lower boundaries within which costs could vary.
The authors posit that a similar approach was used by (i) Ngcamphalala and colleagues, (ii) Östensson and colleagues and (iii) Ginindza and colleagues:
Ngcamphalala C, Östensson E, Ginindza TG. The economic burden of cervical cancer in Eswatini: Societal perspective. PLoS One 2021; 16: e0250113.
Östensson E, Fröberg M, Leval A, et al. Cost of Preventing, Managing, and Treating Human Papillomavirus (HPV)-Related Diseases in Sweden before the Introduction of Quadrivalent HPV Vaccination. PLoS One 2015; 10: e0139062.
Ginindza TG, Sartorius B, Dlamini X, Östensson E. Cost analysis of Human Papillomavirus-related cervical diseases and genital warts in Swaziland. PLoS One 2017;12:e0177762. https://doi.org/10.1371/journal.pone.0177762.
Response 4b: The authors have relooked at this information and can confirm with certainty that cancer stage was taken account of when calculating the total costs for each patient considered. This was done in conjunction with expert guidance, especially since our estimate of average prevalent cases or cases in a single year was low (5). For comparisons the authors varied the average prevalent cases to allow for the inclusion of unrecorded cases. The estimates derived via this approach were also reported.
Further the authors wish to share that this information re: accounting for stage in total costs is reflected under the Treatment parameter which highlights the estimated number of cases in a single year by stage, the direct medical unit cost linked to treatment per stage, and then the total annual direct medical cost per stage. Computation of total costs is done as per equation:
Direct Medical Costs of Disease (mc) = ∑(Mi x Pi)
Where:
Mi is number of cases requiring health care
Pi is the required health care resources unit costs per case
mc is the total costs
On the basis of the aforementioned, we are therefore confident that the estimates presented reflects the total direct medical costs of cervical cancer for 2021.
Comments 5: How was the sensitivity analysis conducted? What parameters were varied during the analysis, and how might these variations affect the results?
Response 5: The authors wish to share that one-way sensitivity analysis was broadly done on the base estimate for each care component. This resulted from multiplying the ‘estimated number of cases in a single year by its corresponding estimated direct medical unit cost. The base estimate was then varied by ±25% to present the lower and upper boundaries derived therefrom.
Further, the authors also varied the following to show how they affect the study results:
- Average number of prevalent cases (cases in a single year) (increased by 50%)
For comparative purposes, the average number of cancer cases in a single year was increased by 50% and the total annual costs reassessed and reported.
- Direct medical unit costs (increased by 50%)
The direct medical unit costs of care components were also increased by 50% and the total annual costs reassessed and reported.
- Treatment costs (lowered by 50%)
Additionally, treatment cost in the original model was reduced by 50%, costs of other parameters remained unchanged and the impact of this manipulation on the total annual costs was assessed and reported.
The results obtained from varying these noted parameters are reflected in both the results section of the manuscript.
|
Kindly note that in addition to the edits done in respect of the comments and/or suggestions of the Reviewer, the authors have made some edits to further improve the article and so as to ensure that there is consistency across all areas of our study re: scope and/or purpose. This included edits to text, figures and tables. |
|
|
|
Thank you |

Round 2
Reviewer 1 Report
Comments and Suggestions for Authors
The responses to the initial review were satisfactory, and I believe this version is more appropriate. The costs have been revised, and the requests have been addressed or justified. However, I consider it necessary to specify in the manuscript title that the estimated costs are associated solely with the treatment of cervical cancer and do not include screening actions or the treatment of precursor lesions.
Additionally, in the discussion section, the authors should consider addressing the limitation of not evaluating the treatment of precursor lesions.
Author Response
|
Comments 1: The responses to the initial review were satisfactory, and I believe this version is more appropriate. The costs have been revised, and the requests have been addressed or justified. However, I consider it necessary to specify in the manuscript title that the estimated costs are associated solely with the treatment of cervical cancer and do not include screening actions or the treatment of precursor lesions.
|
|
Response 1: Thank you for pointing this out. The authors have taken note of this comment and after deliberations have agreed on revising the manuscript title to read:
“Cost Analysis Related to Diagnosis, Treatment and Management of Cervical Cancer in Antigua and Barbuda: A Prevalence-Based Cost-of-Illness Study” Comments 2: Additionally, in the discussion section, the authors should consider addressing the limitation of not evaluating the treatment of precursor lesions. Response 2: Thank you for pointing this out. The authors have taken note of this and wish to indicate that they have since made the necessary edits re: limitations to the manuscript. This is reflected in the highlighted text below as well as in the manuscript.
Lines 474-484 (in manuscript with track changes) or 471-481 (manuscript with accepted track changes) Further, neither our study’s scope allowed for, nor did our estimates reported on the costs of screening, treating or managing precancerous lesions or precursor lesions of cervical cancer. Compared to other studies that included costs for other HPV-related diseases such as CIN, squamous intraepithelial lesions (LSIL and HSIL) and adenocarcinoma in situ (AIS) [6], our presented results could be considered a conservative estimate of the economic burden of cervical cancer. Further studies that may cater to incorporating screening and treatment costs for precursor lesions, comparing screening costs for these lesions with their treatment costs, while also examining positivity rate on cancer outcomes, would be needed to determine the burden of other HPV-related cervical conditions as well as provide a more comprehensive estimate of the economic burden of cervical cancer in Antigua and Barbuda [43,46]. |
|
|
|
Kindly note that in addition to the edits done in respect of the comments and/or suggestions of the Reviewer, the authors have done a review of the manuscript so as to ensure that there is consistency across all areas of our study re: scope and/or purpose. |
|
|
|
Thank you |
